# Large-eddy simulation and stochastic modelling of Lagrangian particles for footprint determination in stable boundary layer

Andrey Glazunov[1], Üllar Rannik[2], Victor Stepanenko[3], Vasily Lykosov[1,3], Mikko Auvinen[2], Timo Vesala[2], and Ivan Mammarella[2]

[1]Institute of Numerical Mathematics RAS , GSP-1, 119991, Gubkina str., 8, Moscow, Russia
[2]Department of Physics, P.O. Box 64, University of Helsinki, 00014 Helsinki, Finland
[3]Moscow State University, Research Computing Center, GSP-1, 119234, Leninskie Gory, 1, bld. 4, Moscow, Russia

*Correspondence to:* A. Glazunov (and.glas@gmail.com)

**Abstract.** Large-eddy simulation (LES) and Lagrangian stochastic modelling of passive particle dispersion were applied to the scalar flux footprint determination in stable atmospheric boundary layer. The sensitivity of the LES results to the spatial resolution and to the parameterizations of small-scale turbulence was investigated. It was shown that the resolved and partially resolved ('subfilter-scale') eddies are mainly responsible for particle dispersion in LES, implying that substantial improvement may be achieved by using recovery of small-scale velocity fluctuations. In LES with the explicit filtering this recovering consists of application of the known inverse filter operator. The footprint functions obtained in LES were compared with the functions calculated with the use of first-order single particle Lagrangian stochastic models (LSM), zeroth-order Lagrangian stochastic models - the random displacement models (RDM) and footprint parameterisations. According to presented LES the source area and footprints in stable boundary layer can be substantially more extended than those predicted by the modern footprint parameterizations and LSMs.

## 1 Introduction

Micrometeorological measurements of vertical turbulent scalar fluxes in the atmospheric boundary layer (ABL) are usually carried out at altitudes $z_M \geq 1.5$ m due to technological limitations of the eddy covariance method. The measurement results are often attributed to the exchange of heat, moisture and gases at the surface. This procedure is not justified for inhomogeneous surfaces because of large area contributing to the flux, and because of variability of the second moments with height. The relationship between the surface flux $F_s(x,y,0)$ and the flux $F_s(x_M, y_M, z_M)$, measured in point $\boldsymbol{x}_M = (x_M, y_M, z_M)$, can be formalized via the footprint function $f_s$:

$$F_s(x_M, y_M, z_M) = \int\limits_{-\infty}^{\infty} \int\limits_{-\infty}^{\infty} f_s(x, y, x_M, y_M, z_M) F_s(x, y, 0) dx dy. \qquad (1)$$

Traditionally, footprint functions $f_s^d(x^d, y^d, \boldsymbol{x}_M) = f_s(x, y, \boldsymbol{x}_M)$ are expressed in local coordinate system with the origin which coincides with the sensor position (here, $x^d = x_M - x$ is the positive upwind distance from the sensor and $y^d = y_M - y$ is the crosswind distance, see Fig. 1a). In horizontally homogenous case these functions do not depend on $x_M$ and $y_M$. In ABL the

surface area contributing to the flux is elongated in wind direction, therefore the crosswind-integrated footprint function $f_s^y$ defined as

$$f_s^y(x^d, z_M) = \int\limits_{-\infty}^{\infty} f_s^d(x^d, y^d, z_M) dy^d, \tag{2}$$

is one of the most required characteristics for the practical use.

5 The measurements of the scalar flux footprint functions in natural environment are restricted (e.g., Finn et al., 1996; Leclerc et al., 1997, 2003; Nicolini et al., 2015) due to the necessity to conduct the emission and detection of artificial tracers. Besides, such measurements are not available for the stably stratified ABL, where the area of the surface influencing the point of measurements increases.

 Modelling approaches used for footprint calculation include stochastic models, such as single particle such as single par-
10 ticle first-order Lagrangian stochastic models based on generalized Langevin equation (LSM) and zeroth-order stochastic models (also known as the random displacement models, RDM) (see the reviews listed in the papers (Wilson and Sawford, 1996), (Wilson, 2015) and the monograph (Thomson and Wilson, 2013)). Besides, one can use the analytical models (e.g., Horst and Weil, 1992; Kormann and Meixner, 2001) and the parameterizations based on the scaling approach (Kljun et al., 2004, 2015). All of these models should be calibrated against the data considered to be representative of real processes. Their
15 results depend on the choice of universal functions in the ABL or in the surface layer (non-dimensional velocity and scalar gradients, non-dimensional dissipation, dispersion of the velocity components etc.). Commonly, the applicability of the analytical models is limited by a "constant flux layer" simplification, assuming that the measurement height $z_M$ is much less than the thickness of the ABL $z_i$. However, under the strongly stable stratification the thickness $z_i$ may be several meters, therefore, the vertical gradients of momentum and scalars fluxes near the surface can be large. It can lead to incorrect functioning of the
20 models designer for, and tested on the data gathered under different conditions.

 Large eddy simulation (LES), employing Eulerian approach for the transport of scalars, was first time applied for a footprint calculation in (Leclerc et al., 1997). Modern computational technologies allow to combine Eulerian and Lagrangian methods for turbulence simulation and particle transport (e.g., Weil et al., 2004; Steinfeld et al., 2008; Cai et al., 2010; Hellsten et al., 2015) and to perform detailed calculations of averaged two-dimensional footprints under different types of stratifications in
25 ABL and footprints $f_s(x, y, \boldsymbol{x}_M)$ over heterogeneous surfaces (for example, urban surface and surfaces with alternating types of vegetation). Some examples of such calculations are given in (Steinfeld et al., 2008; Hellsten et al., 2015).

 Lagrangian transport in LES is complicated by the problem of description of small-scale (unresolved) fluctuations of the particle velocity, which is similar to the problem of subgrid modelling of Eulerian dynamics. A common approach for Lagrangian subgrid modelling in LES is the application of subgrid LSMs (e.g., Weil et al., 2004; Steinfeld et al., 2008; Cai et al., 2010;
30 Shotorban and Mashayek, 2006). This approach requires a number of additional calculations for each particle (e.g., interpolations of subfilter stresses $\tau_{ij}$ and subgrid dissipation $\epsilon$ into the particle position $\boldsymbol{x}^p$). In addition, it is necessary to generate a three-component random noise for each particle, that is a time-consuming computational operation. Numerically stable solution to the generalized Langevin equation (see Sect. 2.3, Eq. (9)) in LES requires a smaller time steps than the steps to solution of Eulerian equations, because local Lagrangian decorrelation time $T_L(\boldsymbol{x}^p, t)$ can be very small.

The statistics of simulated turbulence in LES may significantly differ from the statistics of real turbulence. For example, the use of dissipative numerical schemes or low-order finite-difference schemes usually results in a suppression of fluctuations over almost the entire resolved spectral ranges of discrete models (see e.g., Fig. 16 in Piotrowski et al., 2009). Turbulent fluxes (in the Eulerian representation) associated with these fluctuations are restored by subgrid closure. However, in terms of the Lagrangian transport the effects of distortion of small-scale part of the spectrum are most often not considered.

Numerical simulations of Lagrangian transport in LES are also limited by the low scalability of parallel algorithms. This is due to the impossibility of uniform loading of processors in a joint solution to the Euler and Lagrangian equations, a large number of interprocessor exchanges and unstructured distribution of characteristics required for Lagrangian advection in the computer RAM memory.

Thus, all methods of numerical and analytical determination of the functions $f_s$ have individual drawbacks. At the same time, due to the lack of sufficient amount of experimental data and due to their low accuracy, there are no clear criteria for evaluation of different models.

According to the need of computational cost reduction, one of the objectives of this study is to establish the role of stochastic subgrid modelling in the correct description of the particle dispersion in LES. Is it possible to simplify the calculation and to avoid the introduction of stochastic terms without the loss of accuracy in some integral characteristics, such as the footprints or the concentration of pollutants emitted from the point sources? The role of subgrid fluctuations is reduced with an increase of spatial LES resolution. Therefore, the independence of results from the mesh size is used as a criterion for checking the quality of Lagrangian transport procedures in LES. It will be demonstrated that the subgrid stochastic modelling in LES can be omitted in most cases. Instead, we propose 'computationally cheap' procedure of inverse filtering supplemented by divergent correction of Eulerian velocity to replace the subgrid stochastic modelling in LES (see description below).

Subgrid transport is especially significant near the surface and/or under stable stratification – all are the cases associated with small eddies size. That is why the stable ABL was selected as the key test scenario in this study. We slightly modified the setup of the numerical experiment GABLS (Beare et al., 2006) for this purpose.

LES results are used as the input data for the stochastic models (LSMs and RDMs). These data are pre-adjusted using known universal dependencies and taking into account an incomplete representation of turbulent energy in LES. The comparison of results of different stochastic models and the results from LES allows to specify the parameters for the LSMs and permits to identify the differences between LSMs and RDMs under the conditions which have not been tested previously.

The paper is organized as follows. Section 2 contains the description of some common features of approaches: the implemented numerical algorithm for footprint estimation in LES and LS models (Sect. 2.1); LES governing equations and the definitions of some terminology used for the small-scale modelling description and for the testing of particle transport (Sect. 2.2); the definitions of stochastic models (LSMs and RDMs) and pointing to some problems connected with uncertainty of the choice of turbulent statistics for them (Sect. 2.3 and 2.4). Section 3 contains short description of the numerical algorithms, the turbulent closure for LES model used in this study (Sect. 3.1) and the description of the different approaches for the Lagrangian particles transport in LES tested here (Sect. 3.2). Sect. 4 is mainly devoted to the testing of ability of LES model with rough spatial resolution to reproduce particle dispersion correctly. For this sake, we implemented special setup of the

numerical experiment (see Sect. 4.1) permitting to compare Lagrangian and Eulerian statistics (see Sect. 4.2.2). The focus was made on the approaches with the limited use of subgrid stochastic modelling (see Sect. 4.2.1 where the sensitivity of the computed footprints to the spatial resolution was investigated). The footprints computed with LES model with simple subgrid LSM and RDM (traditional approach) are presented in Sect. 4.2.3 and Sect. 4.2.4. Two-dimensional footprints are shown in

Sect. 4.3. Due to large sensitivity of LSMs to the turbulent statistics we emphasize data preparation for them using LES results, measurements data and similarity laws in Sect. 5.1. Section 5 contains the results of footprint modelling with the use of the set of different RDMs and LSMs (specified in Sect. 5.2) in comparison with LES results (see Sect. 5.3). Section 6 is devoted to the comparison of footprints, computed in LES with the footprint parameterisations based on a scaling approach by Kljun et al. (2004, 2015). Section 7 summarises the results.

In addition to the basic calculation, we carried out a series of tests (see Supplement Sect. S1) under unstable stratification in ABL with different grid steps in LES model. This allows to compare the results presented here with similar results obtained in previous studies (e.g., Steinfeld et al., 2008; Weil et al., 2004) and to verify the performance of our LES model in footprint evaluation. Furthermore, we demonstrate the results of footprint calculations above the inhomogeneous surface (Supplement Sect. S2), with a huge number of particles involved in calculations simultaneously. Computational aspects of technology are

discussed as well.

## 2   Modelling approaches

### 2.1   Numerical evaluation of footprints

Computational methods for determination of footprints often reduce to the implementation of Lagrangian transport of marked particles. Each particle can contain a number of attributes, including its initial coordinate $\boldsymbol{x}_0^p$ and time $t_0^p$. Choose two small

horizontal plates $\delta_S$ and $\delta_M$ for averaging in the neighborhood of zero with the areas $S_S$ and $S_M$, respectively. Define the time interval $T_p = [t_0, t_2]$, during which new particles are ejected near the ground with the intensity $H$ (here $H$ is the mathematical expectation of the new particle number emitted per unit area per unit time) and the interval $T_a = [t_1, t_2]$ $(t_1 > t_0)$, when particles are detected near the point of measurement. If $t_1$ is sufficiently large for the ensemble averaged flux to attain constant value in time, and $T_a$ is quite large for statistically significant averaging, then the footprint $f_s$ can be evaluated by the formula

$$f_s(x_S, y_S, x_M, y_M, z_M) \approx$$

$$\approx \frac{1}{S_M} \frac{1}{T_a} \sum_{p=1}^{n_{SM}} \left( \int\limits_{\delta_S} H(x_0^p + x', y_0^p + y', t_0^p) dx' dy' \right)^{-1} \frac{w^p}{|w^p|} I_{SM}^p, \tag{3}$$

where $n_{SM}$ is the number of intersections of the plane $z = z_M$ by the particle trajectories at horizontal coordinates $\boldsymbol{x}_1^p$ : $(x_1^p - x_M, y_1^p - y_M) \in \delta_M$ in time interval $T_a$, $I_{SM}^p = 1$ if the initial coordinates $\boldsymbol{x}_0^p$ of such particle satisfy the condition $((x_1^p - x_0^p) - (x_M - x_S), (y_1^p - y_0^p) - (y_M - y_S)) \in \delta_S$ and $I_{SM}^p = 0$ otherwise. Here, $w^p$ is the vertical component of the particle velocity at the moment of crossing the plane $z = z_M$. Schematic representation of the algorithm for the footprint

function determination in LES is shown in Fig. 1. In accordance with Eq. (3) and the description above, the particle crossing

the test area $\delta_M$ brings the impact into the value $f_s(x_S, y_S, \boldsymbol{x}_M)$, then the beginning of its modified trajectory shifted in a such way to superpose the point $\boldsymbol{x}_1^p$ with sensor position $\boldsymbol{x}_M$ belongs to the test area $\delta_S$. For example (see, Fig. 1b), when the footprint value is calculated at the point $(x_S, y_S)$ only the red particle is counted, but not the blue particle. Such algorithm of averaging was selected because it permits to refine the footprint resolution on the vicinity of sensor independently on the area of $\delta_M$ using the assumption of some spatial homogeneity.

In the horizontally homogeneous case one can calculate footprint $f_s^d(x^d, y^d, z_M)$ performing averaging over statistically equivalent coordinates of sensor position. For this averaging in LES with periodic domain one can prescribe the coordinates $(x_M, y_M)$ to the domain center and select the area $\delta_S$ to be equal to whole domain size. Analogical methods can be applied when using LSMs or RDMs, whereas in the case of RDMs particle displacement should be used in the Eq. (3) instead of velocity.

Nonuniform Cartesian grid $\boldsymbol{x}_{ij}^d = (x_i^d, y_j^d)$ (where, $-20 \leq i \leq 160$; $-120 \leq j \leq 120$), stretched with the distance from the sensor position, was selected for the footprint functions accumulation in the following sections of this paper. Grid was prescribed as: $(x_0^d, y_0^d) = (0,0)$; $x_i^d = \Delta_{x0} \gamma_x^{|i|} i/|i|$ and $y_i^d = \Delta_{y0} \gamma_y^{|j|} j/|j|$ if $i \neq 0$ and $j \neq 0$; $\Delta_{x0} = \Delta_{y0} = 2$ m; $\gamma_x = \gamma_y = 1.05$. This grid is independent of the LES model resolution and coincides with the footprint grids selected for all runs with LSMs and RDMs.

## 2.2 Lagrangian particles embedded into LES

Lagrangian particle velocity $\boldsymbol{u}^p$ and the particle position $\boldsymbol{x}^p$ can be computed in LES models as follows:

$$u_i^p = \overline{u}_i^{(p)} + u''^{p}_i, \qquad dx_i^p = u_i^p dt. \tag{4}$$

Here $\overline{u}_i^{(p)}$ is the interpolation of the resolved Eulerian velocity into the particle position; $u''^p_i$ are the small-scale unresolved Lagrangian velocity fluctuations associated with Eulerian velocity fluctuations belonging to "subgrid" and "subfilter" scales. Here and later we shall use the designation "subfilter" to denote the fluctuations which belong to the resolved spectral range of the discrete model, but are not reproduced numerically, and the designation "subgrid" for the fluctuations which can not be represented on the grid due to smallness of the scales. LES governing equations for filtered velocity $\overline{\boldsymbol{u}}$ are:

$$\frac{\partial \overline{u}_i}{\partial t} = -\frac{\partial \overline{u}_i \overline{u}_j}{\partial x_j} - \frac{\partial \tau_{ij}}{\partial x_j} - \frac{\partial \overline{p}}{\partial x_i} + \overline{F_i^e}, \qquad \frac{\partial \overline{u}_i}{\partial x_i} = 0, \tag{5}$$

where $F_i^e$ comprises Coriolis and buoyancy forces, $\overline{p}$ is normalized pressure and $\tau_{ij} = \overline{u_i u_j} - \overline{u}_i \, \overline{u}_j$ denotes the modeled "subgrid/subfilter" stress tensor. System of equations (5) can be supplemented by the Eulerian equations of scalars transport:

$$\frac{\partial \overline{s}}{\partial t} = -\overline{u}_i \frac{\partial \overline{s}}{\partial x_i} - \frac{\partial \vartheta_i^s}{\partial x_i} + \overline{Q}_s, \tag{6}$$

where $\overline{Q}_s$ denotes sources intensity; $\vartheta_i^s = \overline{su_i} - \overline{u}_i \, \overline{s}$ are the parameterized "subgrid/subfilter" fluxes. Usually, the fluctuations $\boldsymbol{u}''^p$ are defined to be dependent on some random function $\xi$, introduced in order to provide the missing part of mixing. The particular approaches for computing the unresolved part of particle velocity will be discussed and tested in the following sections.

There is a great practical interest in the calculation of footprints, as well as of spatial and temporal characteristics of pollution transport from localized sources above heterogeneous surfaces and in the areas with complex geometry (in the urban environment, over the surfaces with complex terrain or over the alternating types of vegetation). LES of such flows becomes a routine procedure with increasing performance of computers. However, the calculation of statistical characteristics of Lagrangian trajectories is complicated in this case by the need of transport of huge number of tracers (e.g., Hellsten et al., 2015). For example, it is necessary to calculate the trajectories of about $10^9$ particles (see Supplement Sect. S2) to obtain the footprints above the inhomogeneous surface with the explicitly prescribed obstacles (the task similar to that presented in (Glazunov and Stepanenko, 2015)).

On the other hand, a large number of particles (see, e.g., Supplement Fig.S2.1b) allows to estimate the local instantaneous spatially filtered concentration of the scalar:

$$s_P(\boldsymbol{x},t) = \sum_{p=1,N} G(\boldsymbol{x} - \boldsymbol{x}^p(t)), \tag{7}$$

where $G$ is the function which coincides with the convolution kernel of LES filter operator and $N$ is the total number of particles in the domain. If the mathematical expectation $Q_p$ of a number of new particles ejected in a unit volume during unit time interval is proportional to the Eulerian concentration source strength $Q_p(\boldsymbol{x},t) = C\overline{Q}_s(\boldsymbol{x},t)$, then $s_P(\boldsymbol{x},t) \approx C\overline{s}(\boldsymbol{x},t)$. One can perform the same operations with the "Lagrangian" concentration $s_P(\boldsymbol{x},t)$ as the operations with the Eulerian scalar $\overline{s}$. Below, we will compare the averaged values of $s_P$ and $\overline{s}$ and their spatial variability. Besides, we will use the estimation of concentration $s_P(\boldsymbol{x},t)$ for correcting the particle velocities (see, Sect. 3.2.1, Eqs. (34),(35)), in order to approximate the effect of subgrid turbulence.

## 2.3 Single particle first-order Lagrangian stochastic models (LSM)

Another approach (more widespread due to a lower computational cost) is the replacement of the entire turbulent component of velocity by a random process (Lagrangian stochastic models (LSM)):

$$u_i^p = \left\langle u_i^{(p)} \right\rangle + u_i'^p, \qquad dx_i^p = u_i^p dt. \tag{8}$$

Here $\left\langle u_i^{(p)} \right\rangle$ is the ensemble averaged Eulerian velocity at point $\boldsymbol{x}^p$. Note, that LSMs are assumed to be also applicable under the temporal evolution of turbulence statistics. In this paper we shall consider ABL as it approaches a quasi-steady state. Therefore, due to assumption of ergodicity, ensemble averaging can be replaced by averaging in time and in the directions of spatial homogeneity: $\langle \varphi \rangle \approx \langle \varphi \rangle_{x,y,t}$.

Single particle first-order LSM is formulated as follows. Velocity $u_i'^p$ is described by the stochastic differential equation:

$$du'_i{}^P = a_i(\boldsymbol{x}^p, \boldsymbol{u}^p, t)dt + b_{ij}(\boldsymbol{x}^p, \boldsymbol{u}^p, t)\xi_i^p, \tag{9}$$

where $\xi$ stays for the delta-correlated (usually Gaussian) random noise with the variance $dt$

$$\left\langle \xi_i^p(t)\xi_j^h(t+t') \right\rangle = \delta_{ij}\delta_{ph}\delta(t')dt \tag{10}$$

and with the zero average $\langle \xi_i^p \rangle = 0$; $a_i$, $b_{ij}$ are the functions depending on the Eulerian characteristics of turbulence and on the Lagrangian velocity of the particle. Typically $b_{ij}$ is calculated by the formula

$$b_{ij} = \delta_{ij} \sqrt{C_0 \epsilon}, \tag{11}$$

where, $\epsilon$ denotes the energy dissipation rate, averaged for a fixed coordinate, $C_0$ is the Kolmogorov constant. This kind of random term (arguments are given in (Thomson, 1987) and (Sawford, 1993)) is defined by Lagrangian velocity structure function in the inertial range (see Monin and Yaglom, 1975):

$$D_{ij}(t') = \langle (u_i(t + t') - u_i(t))(u_j(t + t') - u_j(t)) \rangle = \delta_{ij} C_0 \epsilon t' \tag{12}$$

if $\tau_\eta \ll t' \ll T_E$ ($\tau_\eta = (\nu/\epsilon)^{1/2}$ is the Kolmogorov microscale, $T_E = E^2/\epsilon$ is the energy containing turbulent time scale and $E$ is the turbulent kinetic energy.

The function $a_i$ (drift term) determines the behavior of particles at large times $t \sim T_L \sim T_E$ (here $T_L$ is the Lagrangian decorrelation time scale). For spatially inhomogeneous and statistically non-stationary turbulent flows, including ABL, the choice of $a_i$ is usually done according to the well mixed condition (WMC; Thomson, 1987). In general WMC does not lead to a unique solution for $a_i$. Different LSMs are constructed by introducing the additional physical assumptions and can lead to inequivalent results.

Lagrangian models are very sensitive to the choice of universal functions that define the normalized RMS of the vertical velocity $\tilde{\sigma}_w = \langle w'^2 \rangle^{1/2} / U_*$ and non-dimensional dissipation $\tilde{\epsilon} = \epsilon z / U_*^3$ (here $U_*$ is the friction velocity). Besides, the simulation results are affected by the choice of value of $C_0$. It can be shown (e.g., Durbin, 1984; Wilson and Yee, 2007 ) that for one-dimensional LSM, these parameters determine the eddy diffusivity $K_s$ for the scalar in the diffusion limit (when $t \gg T_L$, i.e. at large distances from the source):

$$K_s = \frac{2\sigma_w^4}{C_0 \epsilon} = \frac{2\tilde{\sigma}_w^4}{C_0 \tilde{\epsilon}} U_* z. \tag{13}$$

The data of measurements in the ABL demonstrate large variation. For example, the values of $\tilde{\sigma}_w^2$ range from 1.0 to 3.1 (see Table 1 in Banta et al., 2006). According to Eq. (13) it implies the change of $K_s$ by more than nine times.

There is no consensus on the value of $C_0$ as well. Formally, $C_0$ has the meaning of a universal Kolmogorov constant in Eq. (11). The estimation of this constant for an isotropic turbulence using the data of laboratory measurements and DNS provides an interval $C_0 = 6. \pm 0.5$ (see, Lien and D'Asaro (2002)). However, the values $C_0 \sim 3 - 4$ are often used for LSM of particle transport in ABL independently from the type of the stratification. These values have been obtained by the different methods. For instance, the value $C_0 = 3.1$ for a one-dimensional LSM corresponds to a calibration performed in Wilson et al. (1981) according to observation data Barad (1958); Haugen (1959). This calibration (see, Wilson (2015)) assumes that the turbulent Schmidt number $Sc = K_m/K_s = 0.64$ near the surface (here $K_m$ is the eddy viscosity). It is known that determination of the turbulent Prandtl number $Pr = K_m/K_h$ ($K_h$ - heat transfer eddy diffusivity) and Schmidt number based on observation data is complicated by large statistical errors associated with the problem of self-correlation (Anderson, 2009; Grachev et al., 2007). Therefore, this method of estimation of $C_0$ cannot be considered as final and should be confirmed by future studies. In

Rizza et al. (2010) the values of $C_0$ were determined using the LES-based evaluations of the velocity structure functions and the Lagrangian spectra in convective and neutrally-stratified ABLs. In this study the LES model had relatively low resolution, which can be insufficient for accurate determination of this constant in the inertial subrange (see discussion on the resolution requirements in Lien and D'Asaro (2002)). Nevertheless, the value $C_0 \sim 3$, in the paper by Rizza et al. (2010) is relevant for LSMs applied to the convective ABL, in that case the constant is also responsible for the energy containing time scales which are well resolved in LES. The detailed overview of the methods of determination of the constant $C_0$ can be found in Poggi et al. (2008), where the discussion on the disagreements of the different approaches is also included. The results of the LSMs are very sensitive to the choice for $C_0$ as it was shown earlier by Du et al. (1995), Rotach et al. (1996), Wilson (2015) and many others. Below we show that the commonly used value of $C_0 \sim 3 - 4$ can be greatly underestimated for the use as a parameter in LSMs applied to the stably stratified ABL.

## 2.4 Zeroth-order Lagrangian stochastic models or random displacement models (RDM)

A simplest approach for development of the models of particle dispersion entails replacement of Eulerian advection-diffusion equation

$$\frac{\partial \langle s \rangle}{\partial t} + \langle u_i \rangle \frac{\partial \langle s \rangle}{\partial x_i} = \frac{\partial}{\partial x_i} K_s \frac{\partial \langle s \rangle}{\partial x_i} + Q_s \tag{14}$$

by the stochastic equation for particle position (random displacement models (RDM)):

$$dx_i^p = \langle u_i \rangle \, dt + \frac{\partial K_s}{\partial x_i} dt + \sqrt{2K_s} \xi_i^p. \tag{15}$$

Probability density of particle position $P$ is connected with scalar field concentration $\langle s \rangle$ as follows:

$$\langle s(\boldsymbol{x}, t) \rangle = \int_{R^3} \int_{-\infty}^{t} Q_s(\boldsymbol{x}_0, t_0) P(\boldsymbol{x}, t | \boldsymbol{x}_0, t_0) d^3 \boldsymbol{x}_0 dt_0. \tag{16}$$

Using the Fokker-Planck equation it can be shown that the Eq. (15) is equivalent to the Eq. (14) from the point of view of concentration transport when the time step $dt$ tends to zero (Durbin, 1983; Boughton et al., 1987).

RDM has some major disadvantages. First, it shares the limitation of Eulerian eddy-diffusion treatment of turbulent dispersion, i.e. "K-theory". Correspondingly, it is not able to describe the non-diffusive near field of a source. Also, RDM can not be applied for the convective ABL, where the counter-gradient transport is observed. Besides, it requires the exact values of diffusion coefficient $K_s$, which can not be measured directly.

# 3 Details of LES model used in this study

## 3.1 Numerical algorithms and turbulent closure

System of equations (5 - 6) is discretized using explicit finite-difference scheme with the second-order temporal approximation (Adams-Bashforth method) and fourth-order (fully-conserved for advective terms) spatial approximation of velocity and scalars on staggered grid (Morinishi et al., 1998).

Mixed model (Bardina et al., 1980), expressed as the sum of the Smagorinsky and scale-similarity models, is used for calculation of turbulent stress tensor:

$$\tau_{ij}^{mix} = \tau_{ij}^{smag} + \tau_{ij}^{ssm} = -2(C_s\overline{\Delta})^2|\overline{S}|\overline{S}_{ij} + (\overline{\overline{u}_i\,\overline{u}_j} - \overline{\overline{u}_i}\,\overline{\overline{u}_j}), \tag{17}$$

where $\overline{S}_{ij}$ is the filtered strain rate tensor, $C_s$ is the dynamically determined (Germano et al., 1991) dimensionless coefficient which depends on time and spatial coordinates. The a priori tests using the data of laboratory measurements show that scale-similarity models with Gaussian or box filters provide correlation typically as high as 80% between real and modeled stresses (see overview in Meneveau and Katz, 2000). The significant part of this correlation can be attributed to non-ideality of the spatial filter and use of common information for computing both the real and modeled stresses (Liu et al., 1994). The discrete spatial filter used in this study has a smooth transfer function in spectral space, so it can be supposed that the scale-similarity part of Eq. (17) is mainly responsible for the influence of velocity fluctuations belonging to "subfilter" scales.

The procedure of calculation of the coefficients $X(\boldsymbol{x},t) = (C_s\overline{\Delta})^2$ reduces to minimization of the functional $\Psi(X) = \int_\Omega \varepsilon_{ij}(\boldsymbol{x})\,\varepsilon_{ij}(\boldsymbol{x})d\boldsymbol{x}$ where $\Omega$ is the model domain and $\varepsilon_{ij}(\boldsymbol{x})$ is the the residual of the overdefined system of equations

$$\left(\widehat{XM_{ij}^\tau}\right) - \alpha^2 X(M_{ij}^T) = L_{ij} - H_{ij} + \varepsilon_{ij}, \tag{18}$$

obtained by substitution of mixed model (Eq. 17) into the Germano identity as

$$T_{ij} - \widehat{\tau_{ij}} = \widehat{\overline{u}_i\,\overline{u}_j} - \widehat{\overline{u}_i}\,\widehat{\overline{u}_j}. \tag{19}$$

Here $T_{ij}$ are subgrid/subfilter stresses for the smoothed velocity $\widehat{\overline{\mathbf{u}}}$, obtained by successive application of basic $F_{\overline{\Delta}}$ and test $F_{\widehat{\Delta}}$ spatial filters, $\alpha = \widehat{\overline{\Delta}}/\overline{\Delta}$ is the ratio of the filters widths. Tensors $M_{ij}^T$, $M_{ij}^\tau$, $L_{ij}$ and $H_{ij}$ are calculated as follows:

$$M_{ij}^T = 2\left|\widehat{\overline{S}}\right|\widehat{\overline{S}}_{ij}, \quad M_{ij}^\tau = 2|\overline{S}|\,\overline{S}_{ij},$$
$$L_{ij} = \widehat{\overline{u}_i\,\overline{u}_j} - \widehat{\overline{u}_i}\,\widehat{\overline{u}_j}, \quad H_{ij} = \left(\widehat{\widehat{\overline{u}_i}\,\widehat{\overline{u}_j}} - \widehat{\widehat{\overline{u}_i}}\,\widehat{\widehat{\overline{u}_j}}\right) - \left(\widehat{\overline{u}_i\,\overline{u}_j} - \widehat{\overline{u}_i}\,\widehat{\overline{u}_j}\right). \tag{20}$$

The generalized solution to the discrete analogue of Eq. (18) is searched using the iterative conjugate gradients (CG) method with diagonal preconditioner. To do this, the problem is reduced to a linear system of equations

$$A_\Delta^* A_\Delta X_\Delta = A_\Delta^* R_\Delta, \tag{21}$$

where $X_\Delta$ is the the desired solution (a vector of dimension $N = N_x N_y N_z$ with the values defined in the center of grid cells); $A_\Delta$ and $R_\Delta = L_\Delta - H_\Delta$ are the discrete analogues of the operator and the right hand side of Eq. (18) correspondingly; $A_\Delta^*$ is the transpose matrix. The diagonal preconditioner $P_\Delta$ for CG method was selected as follows:

$$P_\Delta = \left(\alpha^4 M_\Delta^T M_\Delta^{T^*} + \mu(M_\Delta^\tau M_\Delta^{\tau^*} - 2\alpha^2 M_\Delta^T M_\Delta^{\tau^*})\right)^{-1}, \tag{22}$$

where $\mu = const \sim 1$ is the empirical coefficient independent on time and spatial position. The solution $X_\Delta$ contains negative values (unconditional minimization of the functional is used), however, mixed model (Eq. 17) reduces their relative number

compared with the dynamic Smagorinsky model. In the algorithm, negative values are replaced by zeroes. In fact, this dynamic procedure is close to approach proposed in (Ghosal et al., 1995), with the difference that the mixed model was applied here and iterative method was replaced by a faster CG method.

Eddy diffusion models are used for subgrid heat and concentration transfer:

$$\vartheta_i^s = -K_h{}^{subgr} \frac{\partial \overline{s}}{\partial x_i},$$  (23)

here $K_h{}^{subgr} = (1/Sc^{subgr})(C_s\overline{\Delta})^2|\overline{S}|$ is the eddy diffusivity, which is independent on the type of scalar. Subgrid turbulent Schmidt and Prandtl numbers are fixed $Sc^{subgr} = Pr^{subgr} = 0.8$.

A distinctive feature of this model is that the discrete spatial filter operator $F_{\overline{\Delta}} = F_x F_y F_z$ is explicitly involved in calculation of stresses. The following discrete basic filter is selected:

$$F_x(\varphi)_{i,j,k} = (1/8)\varphi_{i-1,j,k} + (3/4)\varphi_{i,j,k} + (1/8)\varphi_{i+1,j,k},$$  (24)

here $i, j, k$ denote a grid cell number, $\varphi$ is any variable. Similar filtering is applied along the coordinates $y$ and $z$. It is reasonable to expect that we get the velocity $\overline{u}$, smoothed according to specified filtering operator as a solution to Eq. (5) supplemented by the mixed closure (Eqs. 17 - 21). Since the discrete filtering operator is invertible, we can find the following velocity at any point and time:

$$u_i{}^* = F_{\overline{\Delta}}^{-1}\overline{u}_i,$$  (25)

which better reflects the small-scale spatial variability. Approximate inverse filter is calculated as a series (Van Cittert, 1931):

$$F_{\overline{\Delta}}^{-1} \approx F_n^{-1} = \sum_{k=0}^{n}(I - F_{\overline{\Delta}})^k,$$  (26)

where $I$ is a unity operator; in the calculations presented below we used $n = 5$. Spatial spectra of "defiltered" velocity $\boldsymbol{u}^*$ under the neutral, unstable and stable stratification were obtained earlier (Glazunov, 2009; Glazunov and Dymnikov, 2013; Glazunov, 2014). It was found in all cases that this procedure improves the small-scale parts of the spectra according to dependence $S \sim k^{-5/3}$, provides better agreement of spectra calculated with the different spatial resolution and improves convergence of non-dimensional spectra if proper length scales are used for normalization.

## 3.2 Methods for Lagrangian particle transport in LES

### 3.2.1 Subgrid and subfiler modelling

Below, the subgrid and subfilter modelling methods used for the simulations in the current study are listed. These methods will be used also in combinations as defined in Sect. 4.2.

### (1) Improvement of Lagrangian transport using inverse filtering of Eulerian velocity field

First, we will use the recovery of "subfilter" fluctuations (Eqs. 25, 26) in order to transport Lagrangian particles more precisely:

$$\boldsymbol{u}^p = \boldsymbol{u}^{*(p)}$$  (27)

Note, that for the use of such a procedure, LES models should exhibit the properties of model with an explicit filtering. Similar approach was recently applied by Michalek et al. (2013) in LES with approximate deconvolution subgrid model (ADM, see Stolz et al., 2001), which can be also considered as the model with explicit filtering. In most cases, the suppression of small-scale fluctuations in LES (particularly in those that use a low-order numerical schemes) occurs as a result of combined effect of approximation errors and the subgrid closure. Therefore, the shapes of effective spatial filters of most models can only be determined by aposteriori analysis of the calculation results.

**(2) Lagrangian stochastic subgrid/subfilter model**

Second, we will apply the subgrid stochastic model proposed in (Shotorban and Mashayek, 2006):

$$du_i^p = \left( -\frac{\partial \overline{p}}{\partial x_i} - \frac{1}{T_L}(u_i^p - \overline{u}_i^{(p)}) \right) dt + \sqrt{C_0 \epsilon} \xi_i^p. \tag{28}$$

The parameter $C_0$ was specified to be equal to 6, because the stochastic part of the model (Eq. 28) is mainly responsible for spatial and time scales in an isotropic inertial subrange of the turbulence. When using dynamic mixed model (Eqs. 17 - 21), a value of $\epsilon$ is not calculated directly, and then it is assumed that the dissipation is locally balanced by shear production and buoyancy production or sink. In addition, since this model can produce a local generation of kinetic energy, the averaging in a horizontal plane was performed to avoid negative values of dissipation:

$$\epsilon = \left\langle -\overline{S}_{ij}\tau_{ij} \right\rangle_{xy} + \frac{g}{\Theta_0} \left\langle \vartheta_3^\Theta \right\rangle_{xy}, \tag{29}$$

where $\vartheta_3^\Theta$ is the vertical subgrid flux of potential temperature and $g/\Theta_0$ is the buoyancy parameter. Time scale $T_L$ was evaluated as:

$$T_L = (E^{subgr} + E^{subf})/\left( \frac{1}{2} + \frac{3}{4}C_0 \right)\epsilon. \tag{30}$$

Thus, the total unresolved kinetic energy was calculated as the sum of "subfilter" energy

$$E^{subf} = \frac{1}{2} \left\langle (u_i^* - \overline{u}_i)^2 \right\rangle_{xy} \tag{31}$$

and "subgrid" energy:

$$E^{subgr} \approx \frac{1}{2} \int\limits_{kmin_i}^{\infty} S_i(k_i)dk_i \approx \frac{3}{4}C_K'\epsilon^{2/3} \sum_{i=1,3} \left( \frac{\pi}{\Delta_{gi}} \right)^{-2/3}. \tag{32}$$

To evaluate the value $E^{subgr}$ it was supposed that "subgrid" fluctuations belong to quite a wide inertial range with the component-wise velocity spectra $S_i(k_i) = C_K'\epsilon^{2/3}k_i^{-5/3}$, and that the minimal wavenumbers for these fluctuations $kmin_i = \pi/\Delta_{gi}$ correspond to wavelengths in two grid steps. Here, $\Delta_{gi}$ is the grid step in the appropriate direction and $C_K' = \frac{18}{55}C_K = 0.5$ is the Kolmogorov constant (here, $C_K \approx 1.5$ is the Kolmogorov constant associated with three-dimensional wavenumbers).

All the values required for a application of this model were linearly interpolated into the particle position everywhere except at heights $z < \Delta_g/2$, where we use the constant values $T_L(\Delta_g/2)$ and $\epsilon(\Delta_g/2)$. This procedure is rather arbitrary, but it does

not have large impact on the results due to the small decorrelation time $T_L(\Delta_g/2)$. Besides, there are no physically grounded reasons for the justification of such interpolations in LES because the resolved velocity in the vicinity of surface is greatly corrupted by the approximation errors. Such procedures should be considered as an adjustments depending on the numerical scheme and on the subgrid closure.

**(3) Random displacement subgrid/subfilter model**

Third, the RDM specified in Sect. 2.4 will be adopted for the Lagrangian particles subgrid dispersion. In this case we shall use the same subgrid diffusivity $K_h{}^{subgr}$ both for the Eulerian scalars (Eq. 23) and for the particles displacement calculations:

$$dx_i^p = \overline{u}_i^{(p)} dt + \frac{\partial K_s^{subgr(p)}}{\partial x_i} dt + \sqrt{2K_s^{subgr(p)}}\, \xi_i^p. \tag{33}$$

This model does not contains the arbitrary specified parameters except those which were already used in the Eulerian LES. The coefficient $K_s^{subgr}$ was linearly interpolated into the particle positions at heights $z \geq z_0$ with the assumption that $K_s^{subgr}(x,y,0) = 0$. A constant value $K_s^{subgr}(x,y,z) = K_s^{subgr}(x,y,z_0)$ was used for $z < z_0$.

**(4) Divergent correction of the Eulerian velocity field**

Finally, in order to find out whether the subgrid mixing is one of the key processes in the dispersion of Lagrangian tracers, we introduced an additional correction to the particle velocities:

$$\boldsymbol{u}_{cor\_div}^{(p)} = \overline{\boldsymbol{u}}^{(p)} + \overline{\boldsymbol{u}}_{div}^{(p)}, \tag{34}$$

where $\overline{\boldsymbol{u}}_{div}$ is the deterministic divergent additive to the velocity field $\overline{\boldsymbol{u}}$:

$$\overline{u}_{div,i} = \frac{\vartheta_i^{sp}}{s_P} \tag{35}$$

with the imposed restriction $\overline{u}_{div,i} = 0$ if $s_P = 0$. Here, the "subgrid" flux $\vartheta_i^{sp}$ is calculated using the same closure as the closure for Eulerian scalars $\overline{s}$, with the only difference that the concentration $s_P$, estimated by the number of particles in a grid cell, is used in Eq. (23).The applicability of this procedure justified because of the large number of particles involved in simulation (in all the cases described below we have at least several dozens of particles in each grid cell).

Correction given by Eqs. (34), (35) does not provide true small-scale mixing, but only introduces an additional "stretching" or "compression" of the small volumes filled with particles and provides concentration fluxes across the borders of grid cells close to "subgrid" fluxes in Eulerian model. Using this correction, we are guaranteed to get a high correlation between the "Eulerian" and "Lagrangian" concentrations (in all our preliminary tests $\langle \overline{s}' s_p' \rangle_{xy} / \sqrt{\langle \overline{s}'^2 \rangle \langle s_p'^2 \rangle} \approx 0.9$).

The idea of such a correction was based on the assumption that details of the mechanism of subgrid mixing have a little influence on the statistics of trajectories at sufficiently large distances from the source and at long enough time $t$. It was assumed that the quick mixing on small spatial scales can be implicitly substituted by the approximation errors arising in the procedures of interpolation and by the errors of discrete solution to the advection equation. Correction brings an additional systematic effect to reduce incorrect particle transport by the large eddies.

### 3.2.2 Simplified velocity interpolation

In preliminary tests it became clear, that trilinear interpolation of each velocity component provides no advantages for footprint calculation in comparison with the following simplified linear interpolation on a staggered grid:

$$
\begin{aligned}
u^{(p)} &= \overline{u}_{i-1/2,j,k} \frac{x_{i+1/2,j,k} - x^p}{\Delta x} + \overline{u}_{i+1/2,j,k} \frac{x^p - x_{i-1/2,j,k}}{\Delta x}, \\
v^{(p)} &= \overline{v}_{i,j-1/2,k} \frac{y_{i,j+1/2,k} - y^p}{\Delta y} + \overline{v}_{i,j+1/2,k} \frac{y^p - y_{i,j-1/2,k}}{\Delta y}, \\
w^{(p)} &= \overline{w}_{i,j,k-1/2} \frac{z_{i,j,k+1/2} - z^p}{\Delta z} + \overline{w}_{i,j,k+1/2} \frac{z^p - z_{i,j,k-1/2}}{\Delta z},
\end{aligned}
\tag{36}
$$

where position $(i,j,k)$ is the center of a grid cell containing the particle. Trilinear interpolation and interpolation given by Eq. (36) provide nearly the same concentration fluxes across the borders of a grid cell, but the latter does not result in additional substantial smoothing of velocity. An exception was made for the grid layer closest to the surface ($z^p < \Delta_g$) where the mean velocity components were adjusted according to the Monin-Obukhov similarity theory with the dimensionless functions taken from (Businger et al., 1971).

## 4 LES of stable ABL and footprint calculations

### 4.1 The setup of numerical experiment

Stable boundary layer at the latitude $73°$ N in almost steady state conditions was considered. The calculations were carried out according to the GABLS scenario (Beare et al., 2006), with the difference that the geostrophic wind $\boldsymbol{U}_g$ has been rotated $35°$ clockwise such that the wind direction near the surface approximately coincides with the axis $x$. The duration of runs is 9 hours. The initial wind velocity coincides with geostrophic velocity $|\boldsymbol{U}_g| = 8$ m/s. The initial potential temperature $\overline{\Theta}$ is equal to the surface temperature $\Theta_s|_{t=0} = 265$ K up to the height 100 m and increases linearly with the rate $d\Theta/dz = 0.05$ K/m if $z > 100$ m. During the calculations, the surface temperature decreases linearly with time: $d\Theta_s/dt = -0.25$ K/hour. Dynamical and thermal roughness parameters $z_0$ and $z_{0\Theta}$ are set to 0.1 m. The calculations were performed at the equidistant grids with steps $\Delta_g = 2.0$ m, 3.125 m, 6.25 m and 12.5 m. The size of the horizontally periodic computational domain was equal to $400 \times 400 \times 400$ m$^3$. The last hour of numerical experiments was used for averaging the results and subsequent analysis.

This setup is based on the observation data (see, Kosoviĉ and Curry (2000)). As it was shown in (Beare et al., 2006), the LES results obtained under the same conditions with the different models converged with the higher grid resolutions. Later, this case was used for testing the LES models e.g. in (Maronga et al., 2015; Zhou and Chow, 2012; Bhaganagar and Debnath, 2015) and many others and for the improvement of subgrid modelling e.g. in (Basu and Porté-Agel, 2006; Zhou and Chow, 2011; Kitamura, 2010). The LES model presented here was tested earlier under the non-modified setup of GABLS in (Glazunov, 2014), where the turbulent statistics above a flat surface and above an urban-like surface were investigated. In all of these studies, LES results were in agreement with the known similarity relationships for the stable ABL. This allows to consider the LES data for GABLS as a reference case for testing of the approaches utilizing the statistical averaging of the turbulence (e.g., see Cuxart et al. (2006), where the intercomparison of single-column models was performed). Several of nondimensional

relationships in stable ABL were collected and presented in (Zilitinkevich et al., 2013). Considered case is also included in the LES database for this study and fits well with the different stability regimes after the appropriate normalization. Therefore, the results obtained in this particular case can be generalized for many cases due to similarity of the stable ABLs. Besides, the presented simulations are easily reproducible and they can be repeated using any LES model which contains the Lagrangian

particle transport routines.

The mean wind velocity and the potential temperature, calculated with the different spatial steps $\Delta_g$, are shown in Fig. 2 The model slightly overestimates the height of the boundary layer at coarse grids, however, the wind velocity near the surface is approximately the same in all runs. As one can see from the Fig.2, the results of simulation are in good agreement with the results from other LES presented in (Beare et al., 2006) (see, http://gabls.metoffice.com for more information). Mean wind

profile computed in accordance with (Högström, 1996) is shown in Fig. 2 by the vertical dashes, in the surface layer part of the domain this "standard" profile for the stable conditions almost coincides with the longitudinal velocity obtained in LES.

Passive Lagrangian tracers were transported simultaneously with the calculations of dynamics. Each particle, when reaching a lateral boundary of domain, is returned from the opposite boundary in accordance with periodic conditions. The reflection condition is used at the ground. The particles are ejected at the height $z_0 = 0.1$ m (one particle per each grid cell adjacent to

surface) with regular time intervals $\Delta t_{ej} = 1$ s. The position of the new particle within a grid cell is set randomly with uniform probability. The ejection of particles takes place continuously from the seventh to the ninth hour of the experiment.

To limit the number of the particles involved in the calculation the absorption condition is applied at the height of 100 meters within ABL. It was verified previously that the upper boundary condition does not have a large impact on the results of calculations of footprints for the heights $z_M$ up to 60 m and for the distances $x - x_M$ considered in this paper (see Appendix A1

and the test with LSM shown by the orange curves in  12). This formulation of numerical experiment allows direct comparison of the concentration of particles $s_P$, estimated by Eq. (7), and the scalar concentration $\overline{s}$, calculated by the Eulerian approach (Eq. 6). For this purpose, additional scalar $\overline{s}$ is calculated from 7-th till 9-th hour with a constant surface flux $F_s = const = 1$, zero initial condition and the Dirichlet condition $\overline{s} = 0$ at the altitude 100 m.

In the last hour of simulation the averaged number of particles in each cell of the grid near the surface was approximately

equal to 700-800, 350-400,180-200 and 110-130 for grids steps $\Delta_g$=12.5 m, 6.25 m, 3.125 m and 2.0 m, respectively. Having such number of particles one can estimate the concentration $s^p(\boldsymbol{x}_{i,j,k}, t_m)$ at each time step, where $\boldsymbol{x}_{i,j,k}$ is the center of a grid cell. It was assumed, that each particle contributes to the concentration $\tilde{s}_P(\boldsymbol{x}_{i,j,k})$ with the weight $r^p_{i,j,k} = (V^p \bigcap V_{i,j,k})/V_{i,j,k}$, where $V^p$ is rectangular neighborhood of its position with the side $\Delta_g$, $(V^p \bigcap V_{i,j,k})$ is the volume of intersection with grid cell, $V_{i,j,k}$ is the cell volume. This averaging is close to the filtering of Eulerian scalar (Eq. 24). The additional normalization

is performed as follows: $s_P = \tilde{s}_P \Delta t_{ej}/\Delta_z$. The concentration $s_P$ corresponds to the number of particles in one cubic meter under the condition that one particle per square meter per second is ejected near the surface. Concentration $s_P$ is numerically equal (excluding errors, determined by different methods of transport) to the concentration of the scalar field $\overline{s}$ if scalar surface flux $F_s = 1$.

Figure  3 shows the resolved and the parameterized components of flux $\langle w's' \rangle$ in runs with different grid steps. It is seen

that the calculation time is not large enough to reach a steady state (the total flux is not constant with the hight, so the average

concentration continues to grow during the last hour). However, it was checked that the flux footprint close to the sensor is not affected by nonstationarity. Besides, we can compare the values of $\overline{s}$ and $s_P$, because the boundary and initial conditions are identical for them.

The unresolved fraction of the flux $F_s^{sbg} = \langle \vartheta_3^s \rangle$ is an essential part of the total flux $F_s^{tot} = \langle \overline{s}\,\overline{w} \rangle + \langle \vartheta_3^s \rangle$. Accordingly, the vertical transport of Lagrangian particles by resolved velocity $\overline{u}$ may be significantly underestimated. Thus, we have "hard" enough test to verify Lagrangian transport in LES with poorly-resolved velocity field.

## 4.2 Sensitivity of LES results on methods of particle transport and spatial resolution

### 4.2.1 Footprint calculation with limited application of subgrid stochastic modelling in LES

Figure 4 shows the scalar flux footprints averaged in crosswind direction $f_s^y(x_M - x, z_M)$ computed by different methods and with different grid steps. In all cases, we have avoided using the subgrid scale stochastic modelling except calculating the velocity of the particles located within the first grid layer $z^p < \Delta_g$. For the curves marked "st_1l", the resultant velocity of the particles near the surface was calculated as follows:

$$\boldsymbol{u}^p = \boldsymbol{u}^{(p)} + r(z^p)\boldsymbol{u}''^p, \tag{37}$$

where the function $r(z^p)$ is defined as $r(z^p) = (1 - z^p/\Delta_g)$ if $z_p < \Delta_g$, $r(z^p) = 0$ $z_p \geq \Delta_g$ and $\boldsymbol{u}''^p$ is the random velocity component, calculated using the stochastic subgrid model (Eq. 28). To take into account the memory effects in Langevin equation, the stochastic model was implemented inside the layer $z^p < 3\Delta_g$, so (because of the smallness of scale $T_L$) this procedure does not lead to significant distortions in the random component of the velocity.

If the particles are advected by the filtered velocity $\overline{u}$ without any correction then the vertical mixing is too weak and the maxima of footprints $f_s^y$ are strongly underestimated and shifted at the large distances from the sensor position. Divergent correction of Eulerian velocity (Eqs. 34, 35) partially improves the results (squares in Fig. 4a,b). For example, maximum of footprint $f_s^y$ for the sensor height $z_M$=30 m (near the fifth computational level) occurs to be close to the maxima of footprints, computed at fine grids, but it is still shifted. Thus, the correction (Eq. 34, 35) alone is not sufficient. Primarily this is due to the weak mixing below the first computational level, where the contribution of the subgrid velocity is crucial.

The inclusion of stochastics within the first layer improves the result (dashed curves in Fig. 4a,b). However, it is not enough to determine footprints at altitudes comparable to the grid spacing.

The advection of particles by the velocity $\boldsymbol{u}^*$ leads to close matching of functions $f_s^y$, calculated with different grid steps (solid lines of different thickness in Fig. 4c,d). The differences between these footprints are not significant from a practical point of view, and can be equally explained by means of the incorrect Lagrangian particles transport, as well as by means of the insufficiently accurate solution to the Eulerian equations on the coarse grid.

### 4.2.2 Spatial variability of scalar concentration inferred by Eulerian and Lagrangian methods

While the particles were advected by the "defiltered" flow we have also used the correction (Eqs. 34, 35). In this case the subgrid diffusion coefficient was reduced twice $K_h^{*subgr} = cK_h^{subgr}$, $c = 0.5$ (coefficient $c = 0.5$ was chosen because about a half of subgrid flux can be restored using "defiltering" : $\langle \overline{s}w^* \rangle - \langle \overline{s}\,\overline{w} \rangle \approx 0.5 \langle \vartheta_3^s \rangle$). We note that when the particles are advected by velocity $\boldsymbol{u}^{*(p)}$, then the presence or absence (crosses in Fig. 4c,d) of correction has no significant effect on the function $f_s^y$. Nevertheless, this procedure may be useful for the following reasons.

In the inertial range of three-dimensional turbulence along with the kinetic energy the variance of a passive scalar concentration is transferred from large scales to small scales with the formation of the spatial spectrum $S_s \sim \epsilon_s \epsilon^{-1/3} k^{-5/3}$ (see (Obukhov, 1949)) (here $\epsilon_s$ is the dissipation rate of the variance of concentration, caused by molecular diffusion). Lagrangian transport of particles by a divergence-free velocity field $\boldsymbol{u}^*$ with the truncated small-scale spectrum is equivalent to Eulerian advection of concentration $s$ without any dissipation. The absence of subgrid-scale part of the velocity spectrum will lead to reduction of the forward cascade and to the accumulation of variance $\sigma_{sp}^2$ in vicinity of the smallest resolved scales.

Figure 5a shows the variances of "Eulerian" concentration $\sigma_s^2(z) = \langle \overline{s}'^2 \rangle_{xyt}$ computed at different grids, and the variances of "Lagrangian" concentration $\sigma_{sp}^2(z) = \langle s_P'^2 \rangle_{xyt}$. One can see that if particles are advected by the velocity $\boldsymbol{u}^{*(p)}$ (crosses), variance $\sigma_{sp}^2$ is much larger than $\sigma_s^2$. If the velocity $\boldsymbol{u}^{*(p)} + \boldsymbol{u}_{div}^{(p)}$ is used (black circles), the values of $\sigma_{sp}^2$ and $\sigma_s^2$ become closer to each other. Besides, the correction (Egs. 34, 35) increases the correlation $\mathrm{corr}(\overline{s}, s_P) = \langle \overline{s}' s_P' \rangle_{xyt}/(\sigma_{sp}\sigma_s)$ of two fields calculated by means of "Eulerian" and "Lagrangian" approaches ( 5b).

One can expect that in more complicated cases (e.g., the turbulent flow around geometric objects and the formation of quasi-periodic eddies) the accumulation of small-scale noise in the concentration field may lead to the incorrect advection of concentration by the resolved eddies. This effect may be also important for inertial particles when the nonphysical variance of concentration can directly affect dynamics. In additional tests it was found that the correction given by Eqs. (34) and (35) prevents particle stagnation in zones with unresolved turbulence during the modelling of urban-like environment. Thus, this correction is desirable for a number of reasons as a practical replacement of subgrid stochastics which requires large computer resources.

### 4.2.3 Particle advection and footprint determination in LES with subgrid LSM

One can obtain footprints close to those presented at Fig. 4 by means of application of the stochastic subgrid model (Eqs. 28-32). The calculations for this model have been carried out at the grids with steps 3.125 m, 6.25 m and 12.5 m (solid lines in Fig. 6a,b). One can note the defect of the stochastic subgrid modelling in LES, which can not be detected by studying of the mean characteristics. In the previous subsection the recovered "subfilter" part of velocity $\boldsymbol{u}'' = \boldsymbol{u}^* - \overline{\boldsymbol{u}}$ and so the subfilter Lagrangian velocity $\boldsymbol{u}''^{(p)}$ were highly correlated with the resolved velocity $\overline{\boldsymbol{u}}$ in time and space. This is due to the specifics of spatial filter (Eq. 24) used for the recovering given by Eqs. (25,26). This filter has a smooth transfer function in spectral space. The analogous effects of non-ideal filters in LES which lead to the high correlations between modelled and measured turbulent stresses were obtained and discussed earlier in Liu et al. (1994) and Meneveau and Katz (2000), where

the laboratory data of turbulent flows were studied. On the contrary, additional mixing in the stochastic model (Eqs. 28-32) is due to random fluctuations, which are not related to $\overline{u}$ strictly. When one uses coarse grids, the energy of these Lagrangian fluctuations should be large enough to restore mixing in vertical direction. This is accompanied by an excessive suppression of the variability of concentration $s_P$ near the surface, where the contribution of subgrid mixing is large (stars in Fig. 5a). The correlation between "Eulerian" and "Lagrangian" concentration is reduced simultaneously (see Fig. 5b). Probably, this defect of employed Lagrangian stochastic model is connected to the horizontal averaging in evaluation of "subgrid" dissipation and energy. Nevertheless, this result shows that in some cases the stochastic subgrid modelling can prevent correct reproduction of the resolved spatial variability of particle concentration in LES along with improvement of the mean transport.

#### 4.2.4 Footprints in LES with subgrid RDM and the comparison of different methods

In Fig. 7 footprints obtained in LES with intermediate resolution $\Delta_g = 6.25$ m are shown. We choose this resolution because LES dynamics is still reproduced sufficiently well, but the effects from the subrgrid/subfilter Lagrangian parametrization are already clearly visible. In addition to the approaches which were already discussed above we applied the subgrid RDM (Eq. 33) and the subgrid RDM in combination with the velocity recovering (Eqs. 25,26) and the correction (Eqs. 34, 35). In the former case we restricted the activity of the subgrid RDM by the multiplying of the diffusivity coefficient $K_h^{subgr(p)}$ in (Eq. 33) on the following ramp function $r(z^p) = (1 - z^p/\Delta_g)$ if $z^p \leq \Delta_g$ and $r(z^p) = 0$ if $z^p > \Delta_g$.

Generally, results are in close agreement with the results of LES with the fine grid except of some details. One can see the intrinsic defect of the RDM when it is applied to the dispersion of particles in near field of a source. Namely, as the RDM is the approximation of the the diffusion process with the infinite speed of the signal prorogation, this model overestimates values of $f_s^y$ in the vicinity of the measurements point location (see Fig. 7d, where this effect is highlighted in the logarithmic scale). Nearly the same effect was obtained in Wilson (2015) (see Figs. 1-3 in its paper, where the footprints from RDM are also shifted left in comparison with the other models). It was also observed that, along with the overestimated vertical mixing, subgrid RDM leads to the propagation of some portion of the particles in the upwind direction (the function $f_s^y(x_M - x, z_M = 10)$ has the small but the positive values if $x_M - x < 0$). In LES with the intermediate resolution the mentioned overestimated mixing exceeds the similar effect in RDM standing alone (see Sect.5.3), because the coefficient $K_h^{subgr}$ is highly variable in time and space and it can attend even larger local values then the magnitude of the averaged turbulent diffusivity $K_h$. At the higher levels of $z_M = 30$ m and $z_M = 60$ m, the footprints are formed as a results of averaging of the turbulent motions over the large spatial distances and over long temporal intervals, and the diffusion approximation becomes to be acceptable. As it will be shown in Sect.5.3, RDM applied alone gives a very close results to the results of LSMs in this particular case of the stable ABL.

In contrast to the subgrid LSM and to the methods of velocity correction proposed above, the advantage of the subgrid RDM consists in the absence of the arbitrary prescribed parameters and in the absence of the need to involve the additional suppositions. In terms of Eulerian statistics, this model is identical to the Eq. (6) (in the limit $dt \to 0$ and with the precision defined by the spatial approximations). From this point of view subgrid RDM can be considered as the "ideal" model, because it is determined by the coefficients which are consistent with LES dynamics of the stratified flow (the same subgrid diffusivity is used for the potential temperature which defines the buoyancy and the interchanges between the kinetic and the available

potential energy). Thus, we have one more confirmation of the validity of the results, except of the invariance with respect to the grid steps.

The impact from the subgrid RDM is reduced when it is applied within the first grid layer only. In this case, the footprints are approximately the same as the footprints computed using the other approaches.

### 4.3 Two-dimensional footprints

The trajectories of large number of particles ($\sim 1.8 \times 10^8$) were simultaneously computed in LES with grid step 2.0 m. Accordingly, one can get statistically grounded estimation of two-dimensional footprint functions $f_s(x - x_M, y - y_M, z_M)$. These functions, computed for the sensor heights $z_M$=10 m and $z_M$=30 m are shown in the Fig. 8a,b. One can see, that the area with the negative values of footprint exists. The negative values of footprints are typical (e.g., Cai et al., 2010; Steinfeld et al., 2008) for the convective boundary layer due to fast upward advection by the narrow thermal plumes and slow downward advection in the surroundings. Here, the negative values of the function $f_s$ are connected to the Ekman spiral and to the mean transport of the particles elevated to large altitudes in the direction perpendicular to the near-surface wind. The negative values of scalar flux footprint show that the vertical turbulent transport of the scalar emitted in the relevant area is basically directed from the upper levels down to the surface. For example, the positive surface concentration flux in this area will lead to negative anomaly of the turbulent flux measured in the sensor position. This does not contradict the diffusion approximation of the turbulent mixing, because mean crosswind advection at the upper levels can produce the positive vertical concentration gradient to the right of near-surface wind.

The contribution of the negative part of the flux to the "measured" flux is significant, as shown in Fig. 8c,d, where cumulative footprints, defined as

$$F(x^d, z_M) = \int\limits_{-\infty}^{x^d} f_s^y(x', z_M) dx', \tag{38}$$

are separated into positive and negative parts $F(x_M - x, z_M) = F^+ + F^-$.

## 5 Stochastic modelling and the comparison with LES

### 5.1 Preparation of turbulence data from LES for LSMs and RDMs

The LES results with grid step $\Delta_g$=2.0 m were used for data preparation. To apply LSM (Eqs. 8, 9) the following Eulerian characteristics are required: the mean wind velocity components $\langle u \rangle$ and $\langle v \rangle$, the second moments $\langle u'_i u'_j \rangle$ and the dissipation $\epsilon$. Stochastic models are even more sensitive to some of these characteristics than the advection of particles in LES. For example, the underestimated values of the turbulent kinetic energy in LES are the consequence of the suppression of small eddies. Nevertheless, these eddies exert relatively small influence on the mixing of scalar, because the effective eddy diffusivity associated with them $K_h^{small} \sim E_{small}^{1/2} l^{small}$ is not large due to small spatial scale. However, the turbulent energy which is substituted into LSM affects results independently of the scale and has to be evaluated with good accuracy.

### 5.1.1 Mean velocity

Mean wind velocity at the height $z_0 < z \le \Delta_g$ was computed using log-linear law:

$$\langle u_i \rangle = U_* \left( \frac{1}{\kappa} ln \left( \frac{z}{z_0} \right) + C_m \frac{z}{L} \right) \times \left. \frac{\langle u_i \rangle}{|\boldsymbol{u}|} \right|_{z=\Delta_g/2}, \quad C_m = 5, \tag{39}$$

and $\langle u_i \rangle = 0$ at $z < z_0$. Here, $U_*$ is the friction velocity, $\kappa = 0.4$ denotes the von Karman constant, $L$ is the Obukhov length at the surface

$$L = -\frac{U_*^3 \Theta_0}{g Q_s}, \tag{40}$$

where $Q_s$ is the kinematic potential temperature flux at the surface, $g = 9.81$ m/s$^2$ is the acceleration of gravity and $\Theta_0 = 263.5$ K is the reference potential temperature (as it was prescribed in presented simulations and in Beare et al. (2006)). Note, that the von Karman constant is not included in the definition of the length $L$ here and later (this alternative definition of the Obukhov length is used along with the traditional one, see e.g. Zilitinkevich et al. (2013) Eq.(41)). The linear interpolation of velocity was used if $z > \Delta_g$.

### 5.1.2 Momentum fluxes

The fluxes $\langle u_i' u_j' \rangle = \langle \overline{u}_i' \overline{u}_j' \rangle + \tau_{ij}^{mix}$ $(i \neq j)$ were interpolated linearly and additionally smoothed everywhere in the domain. These fluxes are shown in Fig. 9a.

### 5.1.3 Variances of velocity components

The variances of velocity components $\sigma_i^2 = \langle u_i'^2 \rangle$ were estimated by formula:

$$\sigma_i^2 = \langle (u_i^{*\prime})^2 \rangle_{x,y,t} + \frac{2}{3} E^{subg}, \tag{41}$$

where $E^{subg}$ is the subgrid energy (Eq. 32) and $\langle (u_i^{*\prime})^2 \rangle$ are the variances of recovered velocity components. The vertical velocity variance has the greatest impact on the functions $f_s^y$. Figure 9b shows the comparison of evaluated normalized RMS $\tilde{\sigma}_w = \sigma_w / |\tau|^{1/2}$ (solid line) with the SHEBA data (symbols; see description in (Grachev et al., 2013, fig. 15b); data kindly provided by Dr A. Grachev). The data are shown in dependence on nondimensional stability parameter $\xi = \kappa z / \Lambda$, where

$$\Lambda(z) = -\frac{|\tau|^{3/2} \Theta_0}{g Q} \tag{42}$$

is the local Obukhov length, determined using values of fluxes of momentum $|\tau|$ and temperature $Q$ at the given height $z$ (local scaling in stable ABL (Nieuwstadt, 1984)). The measurements suggest that the mean value of normalized RMS $\tilde{\sigma}_w \approx 1.33$ if value $\xi$ is small. Figure 9b shows, that our estimation of RMS is slightly less than the measured values in the interval $0.03 < \xi < 0.2$. Respectively, the final values of vertical velocity variance designed for the substitution in stochastic models were corrected as follows: $\sigma_w^2 = 1.33^2 |\tau|$ if $\xi < 1$. At the higher levels the estimation (Eq. 41) was applied.

The final estimations of the variances of velocity components are shown in Fig. 9c by the solid lines. Dashed lines are the filtered resolved velocity $\overline{u}_i$ variances. The estimation of the variance $\sigma_w^2$ using Eq. (41) is shown by the circles. One can see that significant parts of variances were not reproduced explicitly in LES and were recovered using above mentioned assumptions.

### 5.1.4 Turbulent energy dissipation rate

Usual interpolation is not applicable to the calculation of dissipation rate near the surface, where $\epsilon \sim 1/z$. Besides, the values of dissipation $\epsilon_{\Delta k}$ computed in LES at the levels $z_k = (k - 1/2)\Delta_g$ are approximately equal to the averaged values inside the layers $(k-1)\Delta_g < z \leq k\Delta_g$, but not to the physical dissipation at given altitudes. Under the assumption that $|\tau|$ is constant with height and neglecting the stratification inside first layer, one can get the following corrected value of $\epsilon$ at the height $z = \Delta_g/2$:

$$\epsilon|_{z=\Delta_g/2} \approx 2\epsilon_{\Delta 1}/ln(\Delta_g/z_0) \tag{43}$$

Additional analysis showed that, if $z < 0.25z_i$, then the local balance of turbulent kinetic energy (TKE) is well satisfied: $\epsilon \approx S + B$, where $S$ and $B$ are shear and buoyancy production. Therefore, the nondimensional dissipation can be approximated by a formula

$$\tilde{\epsilon} = \frac{\epsilon z}{|\tau|^{3/2}} = \phi_m\left(\frac{z}{\Lambda}\right) - \frac{z}{\Lambda} = \frac{1}{\kappa} + (C_m^\Lambda - 1)\frac{z}{\Lambda}, \tag{44}$$

where

$$\phi_m = \left|\frac{\partial \langle \boldsymbol{u} \rangle}{\partial z}\right| \frac{z}{|\tau|^{1/2}} = \frac{1}{\kappa} + C_m^\Lambda \frac{z}{\Lambda} \tag{45}$$

is the nondimensional velocity gradient; $C_m^\Lambda = 5$, according to the observation data (e.g., Grachev et al., 2013) and LES results (e.g., Glazunov, 2014). Here, the assumption is used that the shear $\partial \langle \boldsymbol{u} \rangle / \partial z$ and the stress $\tau$ are collinear. Previous LES studies of stable ABL (e.g., Beare et al., 2006) also give neglectfully small values of the transport terms in TKE balance. The experimental confirmation of the validity of Eq. (44) can be found in (Grachev et al., 2015), where the dissipation in stable ABL was estimated using the spectral analysis of longitudinal velocity in inertial range. In accordance with this paper: $\tilde{\epsilon} \approx \phi_m$, that is almost indistinguishable from Eq. (44) within the accuracy of the experimental data and the ambiguity of the method of dissipation evaluation.

Discrete values of nondimensional dissipation $\epsilon_{\Delta k} z_k/|\tau|^{3/2}$ are shown in Fig. 10a by circles. Dashed straight line is the universal function (Eq. 44). One can see, that the correction (Eq. 43) makes the dissipation values closer to the function (Eq. 44). Finally, the profile of dissipation $\epsilon_{cf}(z)$ for LSM was corrected as follows (see Fig. 10b). The dissipation was set to be constant below some height $z_e$, and was replaced by universal function $\epsilon = \tilde{\epsilon}|\tau|^{3/2}/z$ up to the level with $z/\Lambda = 1$. The height $z_e$ was chosen in a such way to equalize values of the dissipation averaged in a layer $0 \leq z \leq \Delta_g$ and the dissipation $\epsilon_{\Delta 1}$. Figure 10b shows that the corrected dissipation $\epsilon_{cf}$ (solid line) is very close to "discrete" dissipation $\epsilon_{\Delta k}$ (circles), except for the first computational level.

### 5.1.5 Diffusion coefficients

Random displacement model (Eq. 15) requires the estimation of eddy diffusion coefficient $K_s$. Note, that due to anisotropy, one should use tensor diffusivity $K_s^{ij}$ in a general case. Neglecting this fact, let us assume that the principal axes of the tensor $K_s^{ij}$ are aligned with the coordinate axes. The correspondent coefficients $K_s^{ww}$, $K_s^{uu}$ and $K_s^{vv}$ (see Fig. 9d) can be calculated as follows:

$$K_s^{ww} = -\langle w's' \rangle / \left( \frac{\partial \langle \overline{s} \rangle}{\partial z} \right), \tag{46}$$

$$K_s^{uu} = \frac{\sigma_u^4}{\sigma_w^4} K_s^{ww}, \quad K_s^{vv} = \frac{\sigma_v^4}{\sigma_w^4} K_s^{ww}. \tag{47}$$

The horizontal eddy diffusivities $K_s^{uu}$ and $K_s^{vv}$ are estimated taking into account the expression (13).

One can see that the formula (Eq. 13) provides a good approximation for the coefficient $K_s^{ww}$ if one sets the value $C_0 = 6$. We note, that the data of LES were substantially corrected to get this estimation. Very fine grid simulations are needed to verify and justify the given value. There is no guarantee that this constant is actually universal under different stratification in the ABL.

## 5.2 Specification of LSMs and RDMs tested against LES

The following stochastic models were tested using the data prepared as described above.

**(1)** RDM0 is the random displacements model with uncorrelated components. Particle position is computed by the formula similar to Eq. (15) but with direction-dependent coefficients (see, Eqs. (46), (47) and Fig. 9d). The components of the Gaussian random noise satisfy the Eq. (10).

**(2)** RDM1 differs from RDM0 by using the noise with inter-component correlations:

$$\left\langle \xi_i^p(t)\xi_j^h(t+t') \right\rangle = \frac{\langle u_i' u_j' \rangle}{\sigma_i \sigma_j} \delta_{ph} \delta(t') dt, \tag{48}$$

where $\sigma_i = \left\langle u_i'^2 \right\rangle^{1/2}$.

**(3)** LSM0 is the Lagrangian stochastic model without WMC:

$$du_i'^p = -\frac{u_i'^p}{T_L^i} dt + \sqrt{C_0 \epsilon} \xi_i^p, \quad T_L^i = \frac{2\sigma_i^2}{C_0 \epsilon}. \tag{49}$$

**(4)** LSM1 is based on the one-dimensional well-mixed model:

$$dw^p = \left( -\frac{w^p}{T_L^w} + \frac{1}{2} \frac{\partial \sigma_w^2}{\partial z} \left( 1 + \frac{(w^p)^2}{\sigma_w^2} \right) \right) dt + \sqrt{C_0 \epsilon} \xi_3^p, \quad T_L^w = \frac{2\sigma_w^2}{C_0 \epsilon}, \tag{50}$$

supplemented by uncorrelated horizontal mixing similar to Eq. (49) with the appropriate variances $\sigma_u^2$ and $\sigma_v^2$ .

**(5)** LSMT is three-dimensional Lagrangian stochastic model satisfying WMC, which is proposed by Thomson (1987). For the

incompressible turbulent fluid in a steady state and under the condition of zero mean vertical velocity this model (Thomson, 1987, formula (32)) reads:

$$a_i^p = -\frac{1}{2}\delta_{ij}C_0\epsilon(\tau^{-1})_{ik}u_k'^p + \frac{1}{2}\frac{\partial\tau_{il}}{\partial x_l} + \frac{\partial\left\langle u_i^{(p)}\right\rangle}{\partial x_j}u_j'^p + \frac{1}{2}(\tau^{-1})_{lj}\frac{\partial\tau_{il}}{\partial x_k}u_j'^pu_k'^p,$$

$$du_i'^p = a_i^p dt + \sqrt{C_0\epsilon}\xi_i^p,$$

(51)

where $\tau^{-1}$ is the tensor inverse to the stress tensor.

The setups of numerical experiments with RDMs and LSMs were close to particle advection conditions in LES (absorbtion at the altitude 100 m, ejection at $z_0 = 0.1$m and reflection at $z = 0$). The particles were generated continuously within two hours of modelling. The last hour was used for averaging. The models LSM0 and LSM1 use the value $C_0 = 6$. Three-dimensional model LSMT was applied with $C_0 = 6$ and $C_0 = 8$.

## 5.3   Modelling results

Figure 11 shows crosswind-integrated footprints $f_s^y$ and the corresponding cumulative footprints $F$, computed by LES (bold solid lines, $\Delta_g$=2.0 m) and by stochastic models described above. Footprints are shown for the sensor heights $z_M = 10$, 30 and 60 m.

The models RDM0, RDM1 and LSM1 provide very similar results. Faster mixing is observed in stochastic models below the altitude $z_M = 10$ m in comparison to LES. These differences are not crucial and are compensated in a cumulative footprints

at the distances $x - x_M \sim 1000$ m. The differences can be explained either by insufficient subgrid mixing in LES or by inexact procedure of the data preparation for stochastic modelling. Very weak sensitivity of the models with respect to correlations of particle velocity components is observed as well. Thus, the results close to LES were obtained in stochastic models having the "diffusion limit" with the same or close vertical diffusion coefficient. The significant advantages of LSMs compared to RDMs were not observed in this particular flow.

The substantial disagreements to LES were obtained using three-dimensional Thomson model (Eq. 51) with $C_0 = 6$ and the model LSM0. The last one is designed for the isotropic turbulence and does not satisfy WMC under the conditions considered here. This model leads to overestimated mixing, and such bias does not vanish at large altitudes.

LSMT (Eq. 51) was proposed in (Thomson, 1987) as one of the possible ways to satisfy WMC in three dimensions. In our simulations the error of LSMT is substantial and grows with sensor height. This was shown by Sawford and Guest (1988), who

derived the diffusion limit of Thomson's multidimensional model for Gaussian inhomogeneous turbulence and showed that the implied effective eddy diffusivity for vertical dispersion is:

$$K_s = \frac{2(\sigma_w^4 + \langle u'w'\rangle^2)}{C_0\epsilon}.$$

(52)

Taking into account this expression and Eq. (13) which is valid for the one-dimensional LSM, one can estimate the appropriate value of $C_0$ for LSMT under the conditions considered here: $C_0 \approx 6(1.33^4 + 1)/1.33^4 \approx 8$ (we assume that $\sigma_w/|\langle u'w'\rangle|^{1/2} \approx$

$\sigma_w/|\tau|^{1/2} \approx 1.33$). The results of LSMT with $C_0 = 8$ are in a close agreements with the results of other stochastic models and with the results of LES (open triangles in Fig. 11a,c,e).

Turbulent Prandtl $Pr$ and Schmidt $Sc$ numbers computed using Eulerian approach are shown in Fig. 12a. These numbers coincide and are approximately equal to 0.8 up to the altitude slightly less then 100 m, where the boundary condition for a scalar is applied. Schmidt numbers $Sc$ were calculated also using the concentrations and the fluxes of Lagrangian particles. The models RDM0 and LSM1 provide the values of $Sc$ close to the results of Eulerian model. Calculations by LSMT ($C_0 = 6$)

result in $Sc \approx 0.5 - 0.6$, that is also the sign of the overestimated vertical mixing.

Two-dimensional footprints $f_s(x - x_M, y - y_M, z_M)$, computed by the models RDM0, RDM1 and LSM1 (figures are not shown here) were very close to LES results presented in Fig. 8. In particular, this fact argues that the mechanism of formation of the region with negative values of $f_s$ has a simple nature, which can be easily reproduced in the framework of the diffusion approximation.

The crosswind mixing can be characterized by RMS of transversal coordinates of the particles depending on the mean distance from the source: $Y'^p(X^p) = \langle (y^p - Y^p)^2 \rangle^{1/2}$, where $X^p = \langle x^p \rangle$ and $Y^p = \langle y^p \rangle$ are the mathematical expectations of the particle position. Functions $Y'^p(X^p)$ are shown in Fig. 12b. The models RDM0, RDM1, LSM1 and LSMT (with $C_0 = 6$) result in close horizontal dispersion. All the stochastic models predict slightly less intensive mixing in comparison to LES, that can be a consequence of the inaccurate data preparation algorithm, as well. If one neglects the anisotropy of eddy diffusivity

than this dispersion would be substantially underestimated (see short-dashed line in Fig. 12b, computed by RDM with the coefficients $K_s^{uu} = K_s^{vv} = K_s^{ww}$). One can see, that the choice $C_0 = 8$ in LSMT (open triangles) does not improve its overall performance because the improved vertical mixing is accompanied by the reduced dispersion of particles in the horizontal direction.

Wind direction rotation leads to widening of concentration trace from the point source (see thin dashed line in Fig. 12b,

computed with one-dimensional LSM). At larger distances from the source in the Ekman layer the crosswind dispersion of pollution should be defined by the joint effect of the wind rotation and vertical mixing, but not by the horizontal turbulent mixing.

## 6    Validation of footprint parameterisations based on scaling approach

Footprint parameterisations that are assumed to be valid for a broad range of ABL conditions and measurement heights were

proposed in (Kljun et al., 2004) and recently in (Kljun et al., 2015). These parameterisations are based on a scaling approach. The parameters for these parameterisations were evaluated using backward Lagrangian stochastic particle dispersion model LPDM-B (Kljun et al., 2002). In turn, LPDM-B is based on the forward single particle Lagrangian stochastic model (see (Rotach et al., 1996) and (de Haan and Rotach, 1998)) satisfying WMC. The value of parameter $C_0$ which was selected for LPDM-B stochastic model was set to 3 (see (Kljun et al., 2002)). In parameterisation of LPDM-B, the turbulent statistics and

the wind velocity were assumed to be universal and depend on the surface heat and momentum fluxes, the roughness parameter and the boundary layer height. The exact formulas for all the universal non-dimensional functions under the stable stratification are not presented in (Kljun et al., 2015) and references therein, therefore direct comparison of the turbulence profiles with LES is not possible. Nevertheless, the final approximations (Kljun et al., 2004) and (Kljun et al., 2015) contain the input parameters,

which can be determined from LES: the boundary layer height $z_i \approx 180$ m, Obukhov length $L/\kappa \approx 120$ m, friction velocity $U_* \approx 0.27$ m/s and roughness parameter $z_0 = 0.1$ m. These values were substituted into parameterisations (Kljun et al., 2004) and (Kljun et al., 2015). Fig. 13 shows the comparison of the crosswind-integrated footprint functions $f_s^y$ and cumulative footprints $F$, obtained by different models. The Thomson's model was used with $C_0 = 6, 4$, and 3 for the comparison.

Parametric models provide results which differ substantially from all the abovementioned approaches. Both of the models (Kljun et al., 2004) and (Kljun et al., 2015) predict faster mixing. One can see, that LSMT, which is itself too dispersive in comparison with 1-D LSMs and RDMs, does not reach the values predicted by parameterisations from (Kljun et al., 2004) and (Kljun et al., 2015), even if one chooses the smaller values of $C_0$. It means, that parameterisations of turbulence profiles must have significant impact and are one of the reasons for deviation between models from (Kljun et al., 2004) and (Kljun et al., 2015) and LES. Finally, it can be seen from the Fig. 13, that the top boundary condition (absorbtion of particles at the height 100 m) does not affect the footprints obtained in LSMT.

## 7    Conclusions

Scalar dispersion and flux footprint functions within the stable atmospheric boundary layer were studied by means of LES and stochastic particle dispersion modelling.

It follows from LES results that the main impact on the particle dispersion can be attributed to the advection of particles by resolved and partially resolved "subfilter-scale" eddies. It ensures the possibility to improve the results of particles advection in discrete LES by the use of recovering of small-scale partially resolved velocity fluctuations. If one uses the LES model with the explicit filtering, then this recovering is straightforward and consists of application of the known inverse filter operator. Apparently, a similar method can be implemented for other LES when the spatial filter is not specified in an explicit form. This would require, however, the prior analysis of the modeled spectra to identify an effective spatial resolution and the actual shape of the implicit filter. For substantial improvement of particle transport statistics, it is enough to use subgrid Lagrangian stochastic model within the first computational layer only, where LES model becomes equivalent to simplified RANS-model.

When the particles are advected by a divergence-free turbulent velocity field, then the variance of the particle concentration can be accumulated at small spatial scales. In the considered case, it does not affect directly the particle advection by the large eddies and gives no significant influence on the results of footprint calculations. In those cases, when the instantaneous characteristics of the scalar field of particle concentration are important, the additional correction to particles velocities may be required. It can be done both through the introduction of stochastics, resulting in the diffusion of concentration, and through the "computationally inexpensive" divergent correction of the Eulerian velocity field.

Under the stable stratification, to calculate the flux footprint, it is preferable to use stochastic models, which describe the particle dispersion close to the process of scalar concentration diffusion with the effective coefficient $K_s^{ww}(z) = - \langle w's' \rangle / (\langle ds \rangle / dz)$ in a vertical direction. RDM and one-dimensional "well-mixed" LSM tested in this study are the examples of such stochastic models. The optimal value for the parameter $C_0$ for LSMs is found to be close to 6 under the conditions considered here. This value coincides with the estimation of Kolmogorov Lagrangian constant in isotropic homogeneous turbulence. It provides addi-

tional justification for use of LSMs in stable ABL, due extending their of its applicability over a wider range of scales including the inertial subrange. Stochastic models that use smaller values $C_0 \approx 3 - 4$ (this choice is widespread now) may produce extra mixing and the shorter footprints, respectively. Note that the estimation $C_0 = 6$ is based on the LES results combined with the SHEBA data (Grachev et al., 2013), where the nondimensional vertical velocity RMS was evaluated as $\tilde{\sigma}_w \approx 1.33$ (the exact estimation of this value in LES is restricted by the resolution requirements). In the cases when LSMs utilize smaller values of $\tilde{\sigma}_w$ the parameter $C_0$ should be reduced accordingly (for example, $C_0 \approx 4.7$ will be the best suited parameter for LSMs with the widely used value $\tilde{\sigma}_w \approx 1.25$ prescribed).

One-dimensional stochastic models can be supplemented by the horizontal particle dispersion in a simple way. Introduction of the correlation between particle displacement components in RDM does not improve or change results substantially. However, the coefficients of horizontal diffusion $K_s^{uu}$ and $K_s^{vv}$ for RDMs can be evaluated through the vertical diffusion coefficient $K_s^{ww}$ multiplied by the square of velocity component variances ratio.

Model LSM1, constructed as a combination of independent stochastic models in each direction (well-mixed in the vertical direction only) gives reasonable results although this model does not satisfy WMC in general. In contrast, the three-dimensional Thomson model with WMC and $C_0 = 6$ provides overestimated vertical mixing, which is manifested in a too small Schmidt number values and in reduced lengths of the footprints. Thomson model with $C_0 = 8$ produces true mixing in vertical direction, but underestimates the mixing in crosswind direction.

Accordingly, one can recommend another well-mixed stochastic model proposed in (Kurbanmuradov and Sabelfeld, 2000). It was developed under the assumption that the vertical drift term does not depend on the horizontal velocity components, and the vertical component of this model coincides with LSM1. Prior to use, this model should be modified in an appropriate way to take into account the variation of momentum fluxes with height.

According to presented LES, the source area and footprints in stable ABL can be substantially more extended than those predicted by the modern footprint parameterisations and LSMs. The following reasons were identified in this study: 1) too small values of the parameter $C_0$ are used; 2) the possible overestimated vertical mixing provided by some stochastic models based on well-mixed condition; 3) universal functions for turbulent statistics that are likely to cause additional deviation in the case of stable turbulent Ekman boundary layer studied here.

## 8   Code availability/Data availability

The code of LES model is available by request for the scientific researches in cooperation with first author (and.glas@gmail.com). The data from LES are attached to the supplement. These data were prepared as it was discussed in Sect. 5.1 and can be used for the stochastic models evaluation. Besides, supplement contains the data for crosswind-integrated footprints and two-dimensional footprints obtained in LES (see Fig.6 and Fig.9).

**Appendix A: Assessing the influence of the artificial top boundary condition on the LES results**

To confirm the small impact of the top boundary condition on the results presented above, an additional run was performed (LES with $\Delta_g = 6.25$ m and subgrid LSM, see Sect. 3.2.1(2)). The setup of this numerical experiment was identical to those described in Sect. 4.1, but all particles were retained inside the LES model domain after their ejection (reflection condition was prescribed at the top of the domain). The footprint functions $f_s^y$ obtained in this run are shown by blue curves in Fig.A1a,b,c. The footprints from the particles, which attained the level $z = 100$ m at least one time (the particles were marked by the special identifier in numerical code), were also evaluated (see, dashed red lines in Fig.A1a,b,c). For comparison the footprints with the applied absorbtion of the particles at the level $z = 100$ m are shown by the green lines and the crosses in Fig.A1a,b,c. One can see, that the impact from the particles which were returned from the levels above 100 m is neglectfully small for the sensor heights $z_M = 10$ m and $z_M = 30$ m. For the level $z_M = 60$ m, the influence of the artificial boundary condition is visible beginning from the distances $x_M - x > 6$ km.

The functions $f_s^y(x_M - x, z_M, t_1, t_2)$ are presented in Fig. A1d. Here, $[t_1, t_2]$ is the interval of the time averaging (see, Sect.2.1), shown in the legend in seconds (here, $t_1$, $t_2$ is the time starting from the beginning of the particle ejection). One can see, that the footprints are developed sequentially, the fast and the intensive processes form the footprint function peak first, and it remains to be unchanged later. Figure A1d is included with the aim to demonstrate, that the shape and the value of the footprint function within a large enough range of the distances $x_M - x$ can be independent of the total vertical scalar flux value. The normalized vertical fluxes $\langle F_s(z_M)/F_s(0) \rangle_{[t_1, t_2]}$ are shown also and they grow approximately twice, depending on the time averaging interval.

Finally, we want to mention that this is very specific, and it may be different for different types of ABL. We select the described setup of the numerical experiment intentionally for the sake of convenience of the comparisons of statistics obtained by the Eulerian and the Lagrangian methods. This provides additional ability for the testing of Lagrangian particle transport routines implemented in the LES model code.

*Acknowledgements.* This research is implemented in framework of Russian-Finnish collaboration, funded within CarLac (Academy of Finland, 1281196) and GHG-Lake projects. Russian co-authors are partially supported by Russian Foundation for Basic Research (RFBR 14-05-91752, 15-05-03911 and 16-05-01094). Finnish co-authors acknowledges also the EU project InGOS, and National Centre of Excellence (272041), ICOS-FINLAND (281255), Academy professor projects (1284701 and 1282842) by Academy of Finland.

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

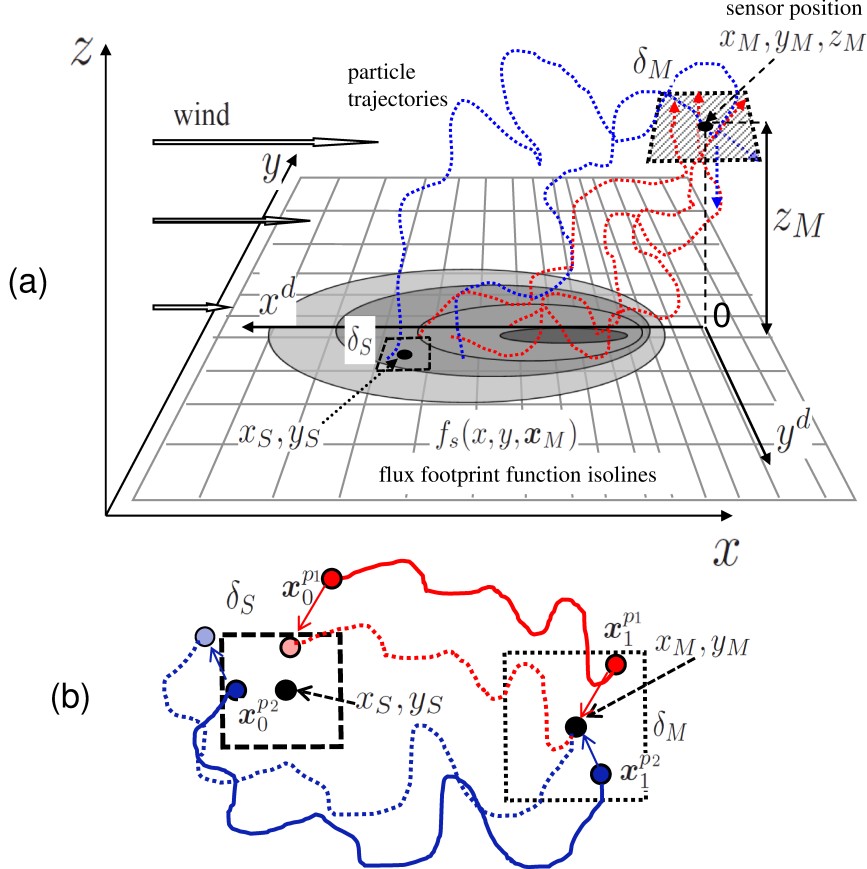

**Figure 1.** Schematic representation of footprint evaluation algorithm. (a) Setup of numerical experiment. (b) Example of two trajectories (red and blue bold curves). Shifted trajectories are shown by the dashed lines. Particle brings the impact into the value $f_s(x_S, y_S, \boldsymbol{x}_M)$ if it intersects the test area $\delta_M$ in vicinity of the sensor position $\boldsymbol{x}_M$ and the origin of modified trajectory belongs to the test area $\delta_S$.

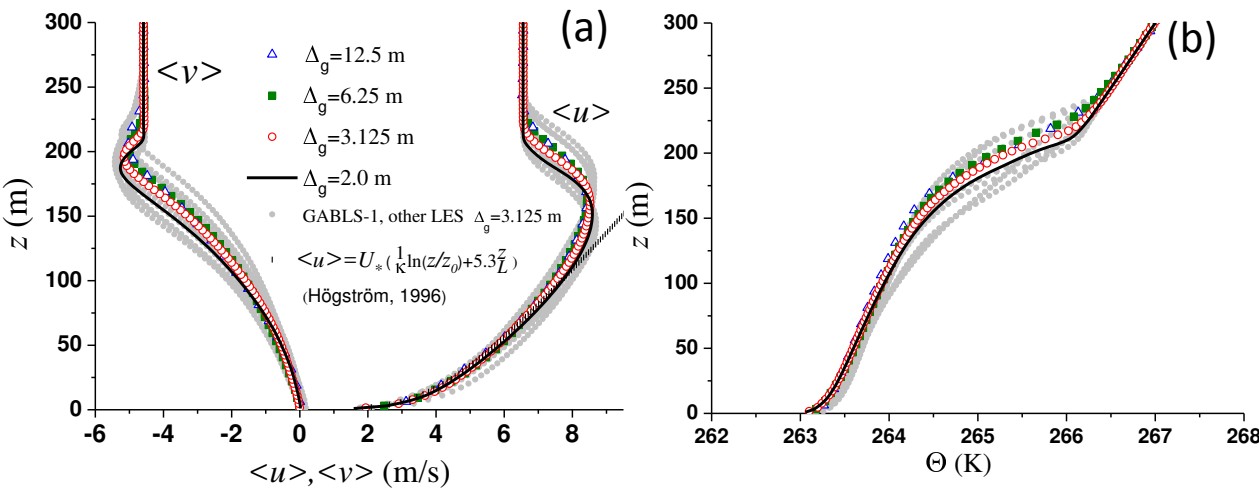

**Figure 2.** Mean wind velocity $\langle u \rangle$ (a) and temperature $\langle \Theta \rangle$ (b) in runs with different grid steps (spatial step is pointed in legend). Gray dots are the data from other LES models obtained in (Beare et al., 2006) (wind velocity is rotated $35^o$ clockwise). 'Standard' wind profile for stable conditions in accordance with Högström (1996) is shown by the vertical dashes.

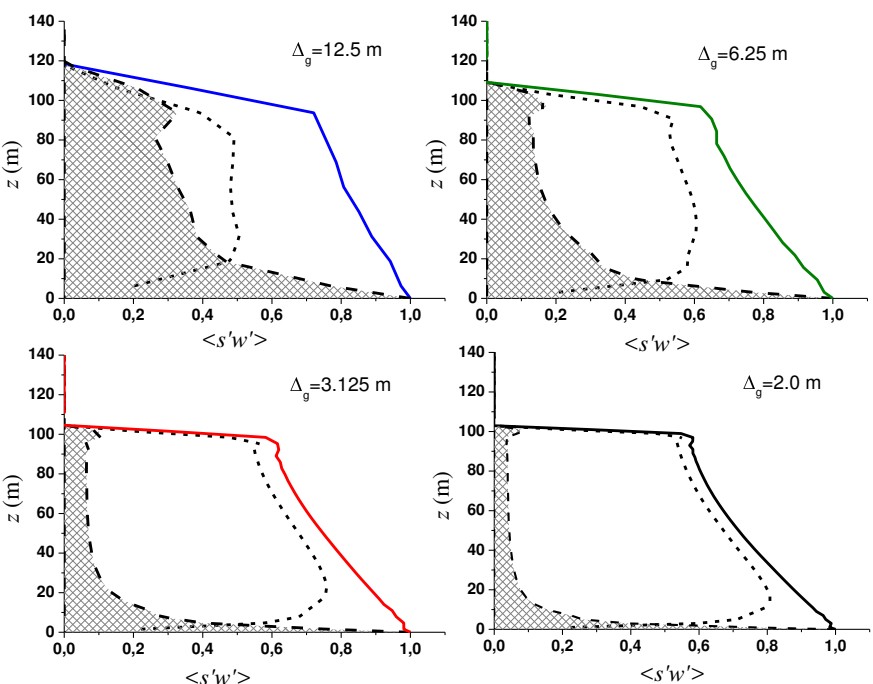

**Figure 3.** Total $F_s^{tot} = \langle \overline{s}\,\overline{w} \rangle + \langle \vartheta_3^s \rangle$ (solid lines), resolved $F_s^{res} = \langle \overline{s}\,\overline{w} \rangle$ (short-dashed lines) and "subgrid" $F_s^{sbg} = \langle \vartheta_3^s \rangle$ (long-dashed lines with shading) scalar fluxes in the runs with different grid steps $\Delta_g$.

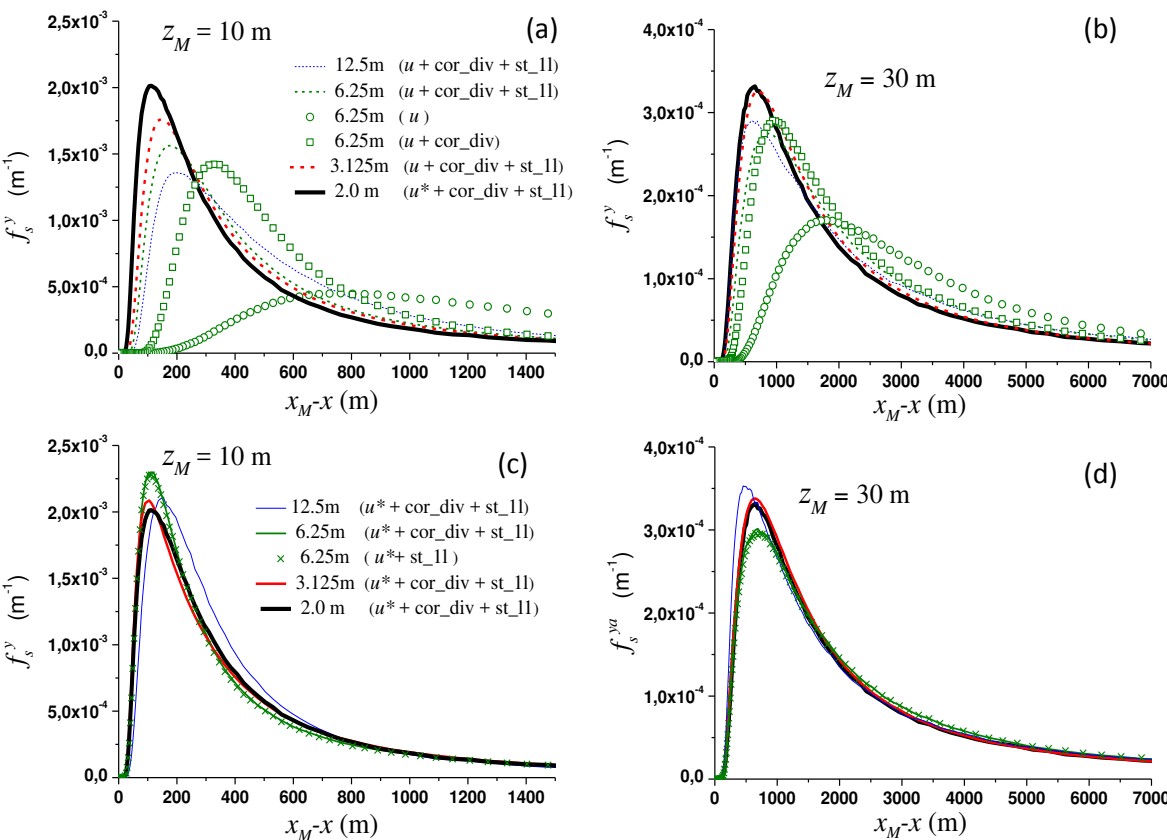

**Figure 4.** Crosswind-integrated scalar flux footprints $f_s^y$ in stable ABL, computed by different methods and with different grid steps; (a,c) sensor height $z_M$=10 m, (b,d) $z_M$=30 m. Grid steps and methods are indicated in the legend: $u$ - particles are transported by a filtered LES velocity $\overline{u}$; $u^*$ - particles are transported by recovered velocity $u^* = F^{-1}\overline{u}$; cor_div - the additional correction of velocity (Eqs. 34, 35); st_ll - stochastic subgrid model (Eq. 28) is applied for the particles within the first computational grid layer.

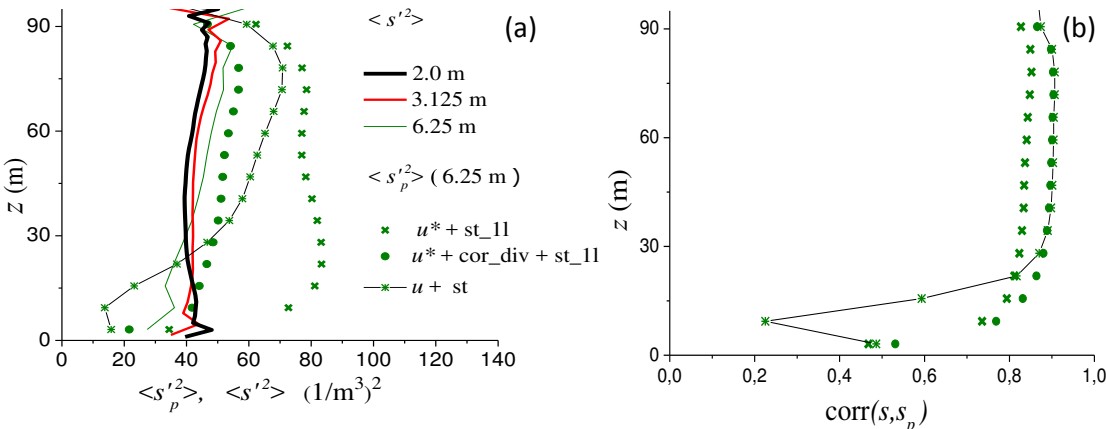

**Figure 5.** (a) Variance $\sigma_s^2 = \left\langle \overline{s}'^2 \right\rangle$ of concentration of Eulerian scalar (solid lines) and variance $\sigma_{sp}^2 = \left\langle s_P{}'^2 \right\rangle$ of concentration $s_P$, determined by Lagrangian particles (symbols); grid steps and the methods of calculations are shown in legend, symbolic notations are the same as in Fig. 4; stars - stochastic model (Eqs. 28-32) is used throughout domain. (b) Correlation $\mathrm{corr}(\overline{s}, s_P) = \left\langle \overline{s}'s_P' \right\rangle_{xyt} / (\sigma_{sp}\sigma_s)$ between "Eulerian" and "Lagrangian" concentrations. For remaining notations see the caption of Fig. 4.

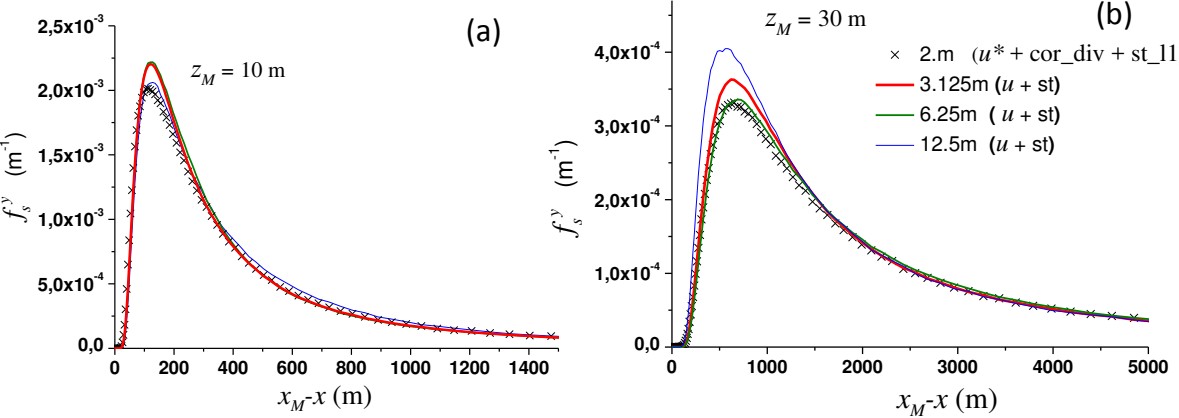

**Figure 6.** Crosswind-integrated scalar flux footprints $f_s^y$, computed using stochastic subgrid model (Eq. 28-32); (a) sensor height $z_M{=}10$ m, (b) $z_M{=}30$ m. Grid steps are given in the legend. Crosses denote footprints computed with subgrid LSM applied for the particles within the first grid layer only.

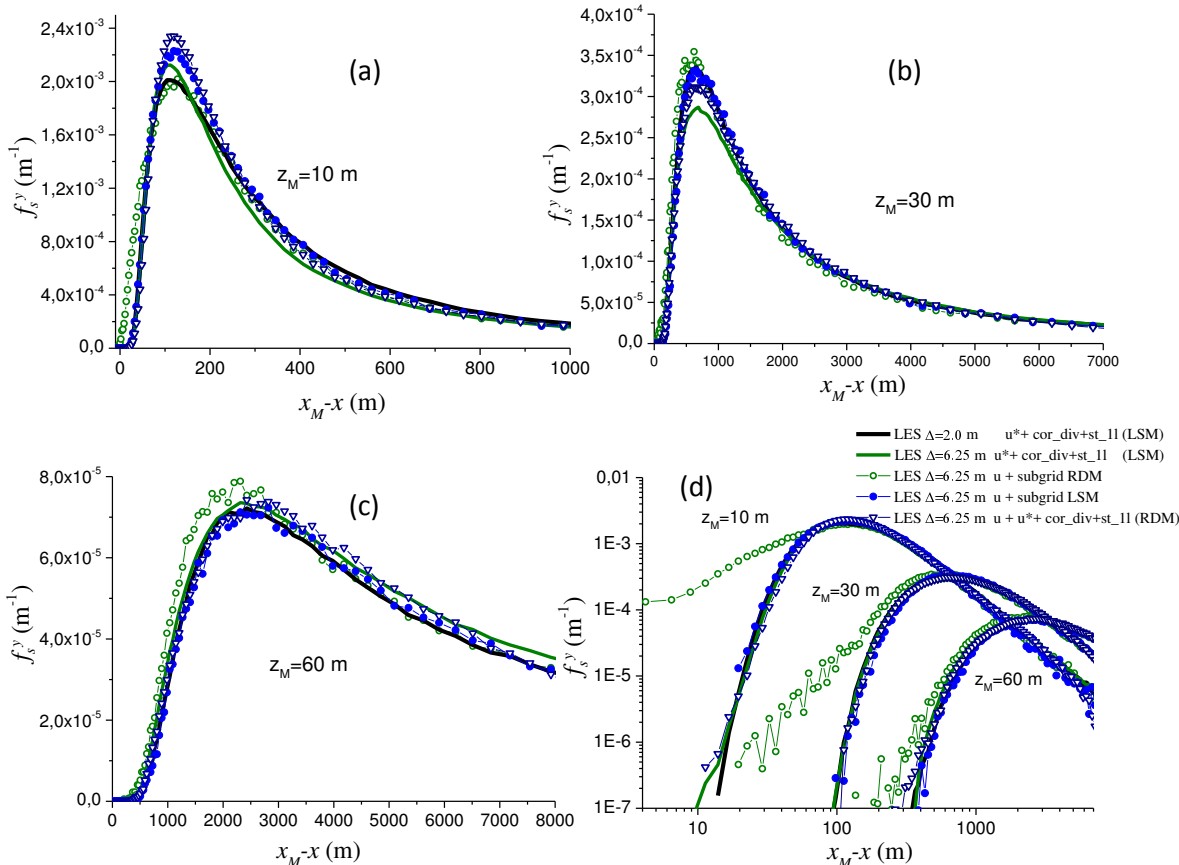

**Figure 7.** Crosswind-integrated scalar flux footprints $f_s^y$, obtained in LES with $\Delta_g = 6.25$ m using different stochastic Lagrangian subgrid models RDM (Eq. 33) and LSM (Eqs. 28-32); The results obtained with these subgrid models applied within the first computational grid layer in combination with velocity recovering $\boldsymbol{u}^* = F^{-1}\overline{\boldsymbol{u}}$ and correction of velocity (Eqs. 34, 35) are also shown. Black lines are the footprints in LES with $\Delta_g = 2.0$ m.

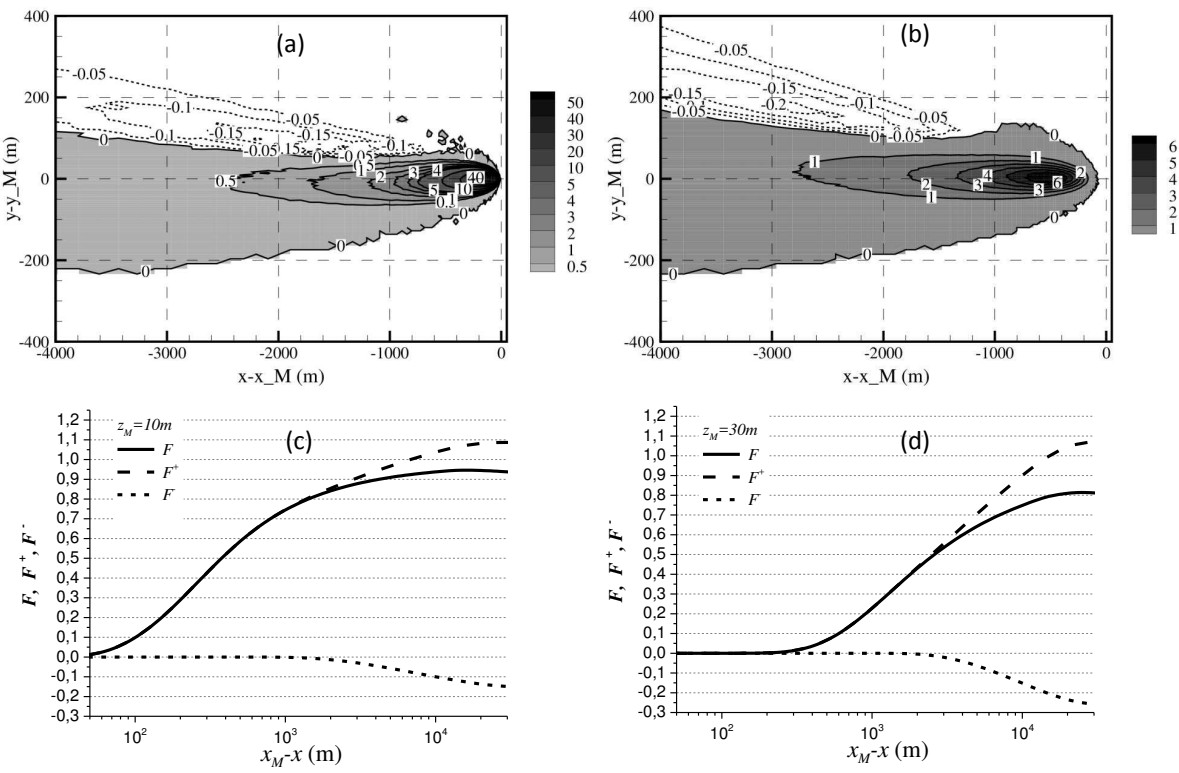

**Figure 8.** Two-dimensional footprints $f_s(x - x_M, y - y_M, z_M)$ $(\times 10^{-6} \text{m}^{-2})$ for sensor height $z_M$=10 m (a) and $z_M$=30 m (b) and the corresponding crosswind-integrated cumulative footprints $F(x_M - x)$ (c) and (d); long dashed line - $F^+$ (impact of the area with positive values of $f_s$); short dashed line - $F^-$ (impact of area with negative values).

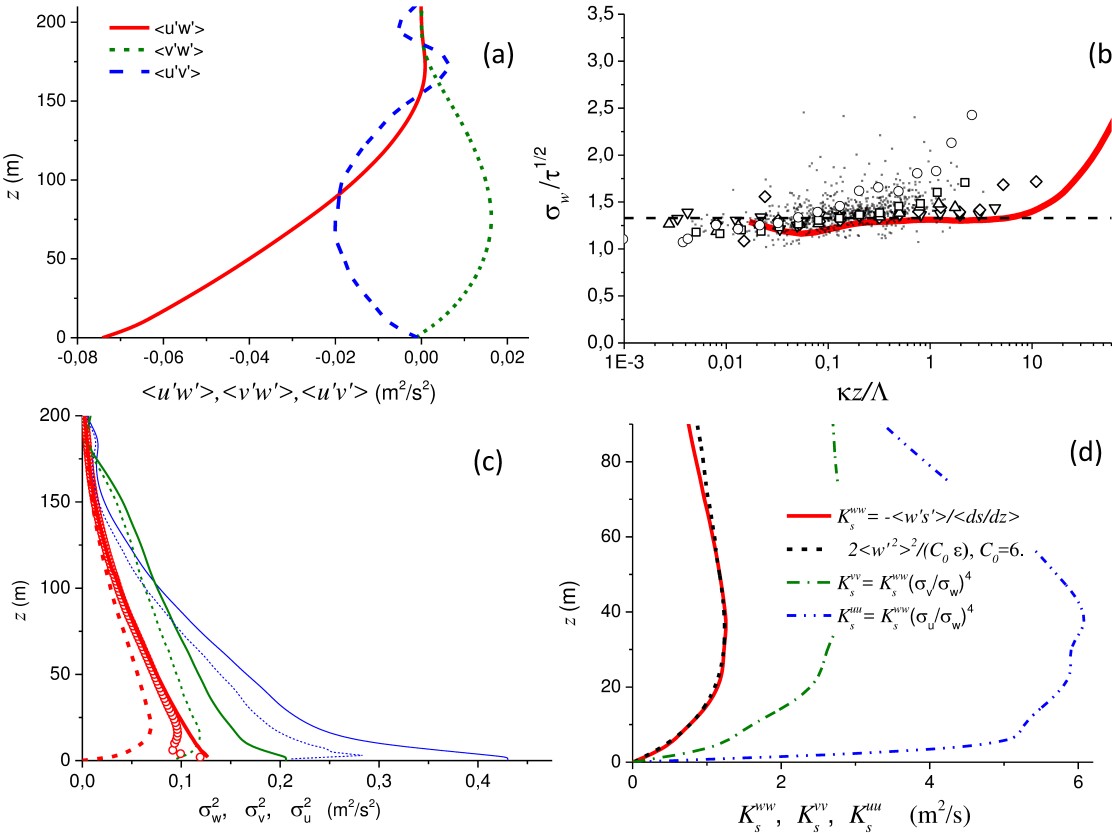

**Figure 9.** (a) Total momentum fluxes obtained in LES with $\Delta_g$=2.0 m. (b) Normalized RMS of vertical velocity $\tilde{\sigma}_w = \sigma_w/|\tau|^{1/2}$ depending on a dimensionless parameter $z/\Lambda$ (solid red line - estimation using LES data $\sigma_w = (\langle w^{*2}\rangle + 2/3E_{subgr})^{1/2}$; symbols - measurements (Grachev et al., 2013) at different heights). (c) Variances of velocity components (dashed line - resolved fluctuation; solid lines - the final estimation for LSM; bold red lines - vertical component, green curves of medium thickness - crosswind component, blue thin lines - longitudinal component, circles - evaluation of $\sigma_w^2$ by Eq. (41)). (d) Vertical effective eddy diffusivity $K_s^{ww}$ (red solid line - coefficient calculated by the gradient and flux of scalar; dashed line - estimation of coefficient using Eq. (13) with $C_0 = 6$); estimations of diffusion coefficients in crosswind direction $K_s^{vv}$ (green dash-dot line) and coefficient in longitudinal direction $K_s^{uu}$ (blue dash-dot-dot line).

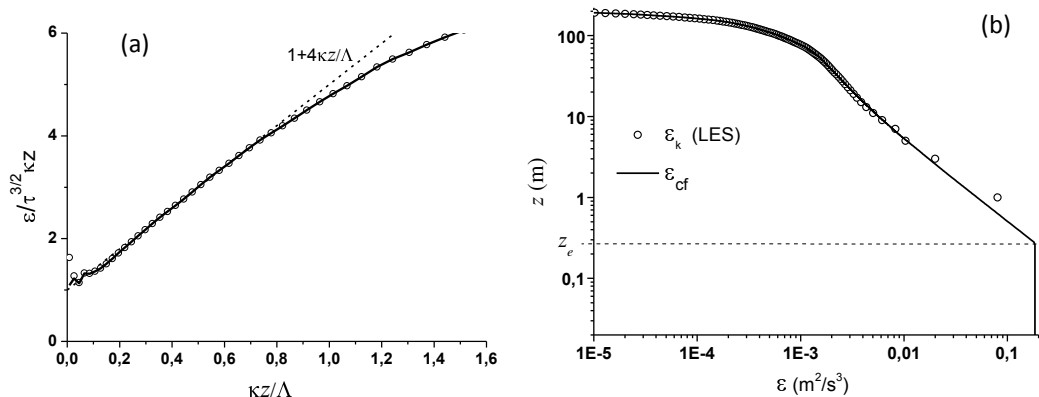

**Figure 10.** (a) Discrete (LES) nondimensional dissipation $\epsilon_{\Delta k} \kappa z_k / |\tau|^{3/2}$ (circles), corrected values (solid line), universal function (Eq. 44) (dashed straight line). (b) Simulated discrete dissipation $\epsilon_{\Delta k}$ (circles) and corrected dissipation $\epsilon_{cf}(z)$ for LSM (solid line). Dashed horizontal line denotes the height $z_e$, which was chosen in order to equalize the integral values of the corrected dissipation and the discrete dissipation.

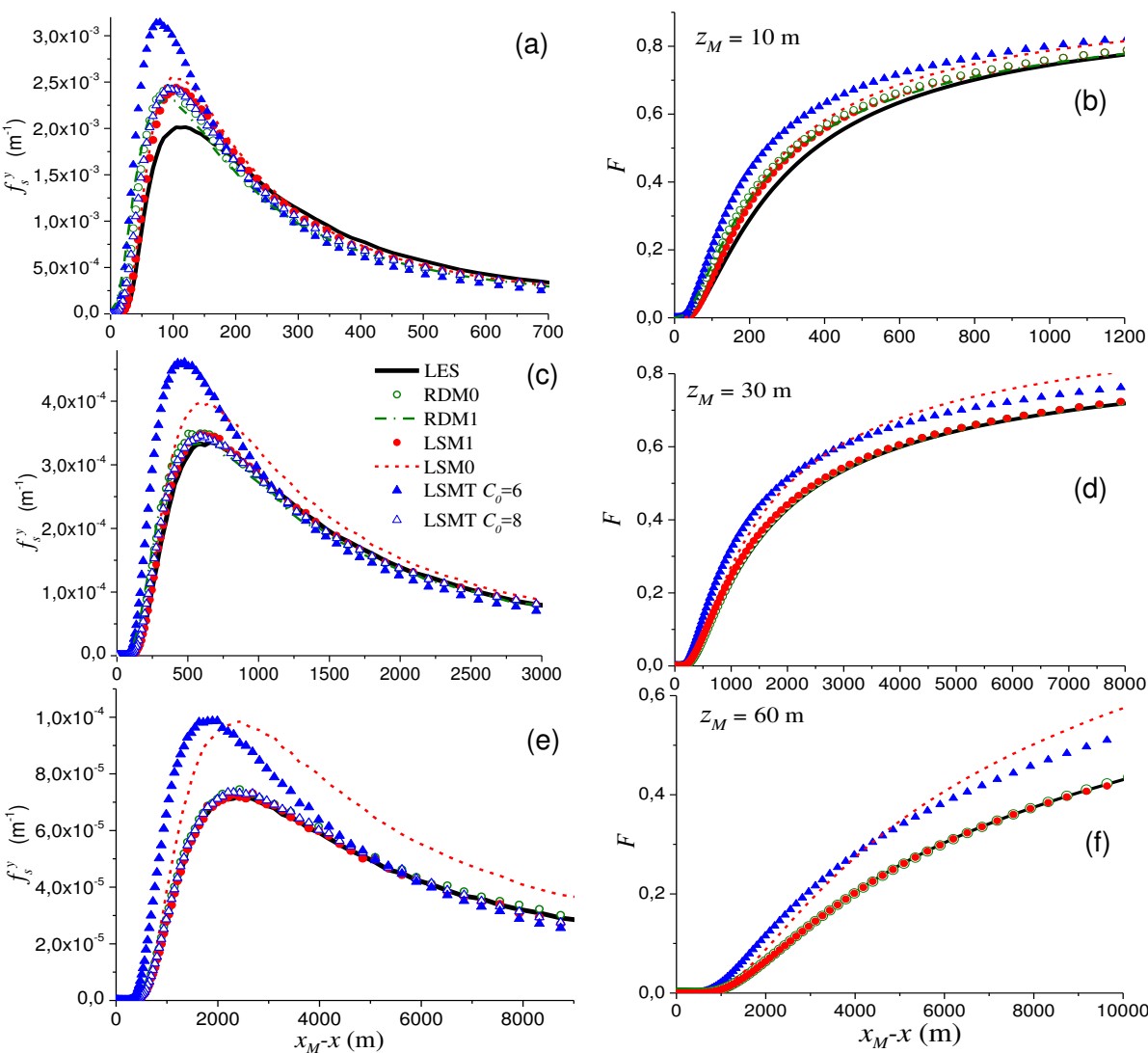

**Figure 11.** Crosswind-integrated footprints $f_s^y$ (a,c,e) and cumulative footprints $F$ (b,d,f) for sensor height $z_M = 10$ m (a,b), $z_M = 30$ m (c,d) and $z_M = 60$ m (e,f). Solid lines - LES with grid steps $\Delta_g$=2.0 m. Blue triangles - LSMT (Thomson, 1987) with $C_0 = 6$, open triangles - LSMT with $C_0 = 8$. Short-dashed line - LSM0 (Lagranian stochastic model without well-mixed condition). Red circles - LSM1 (LSM with WMC for vertical mixing). Open green circles - RDM0 (uncorrelated random displacements model). Dash-dot green line - RDM1 (random displacements model with correlation between displacement components).

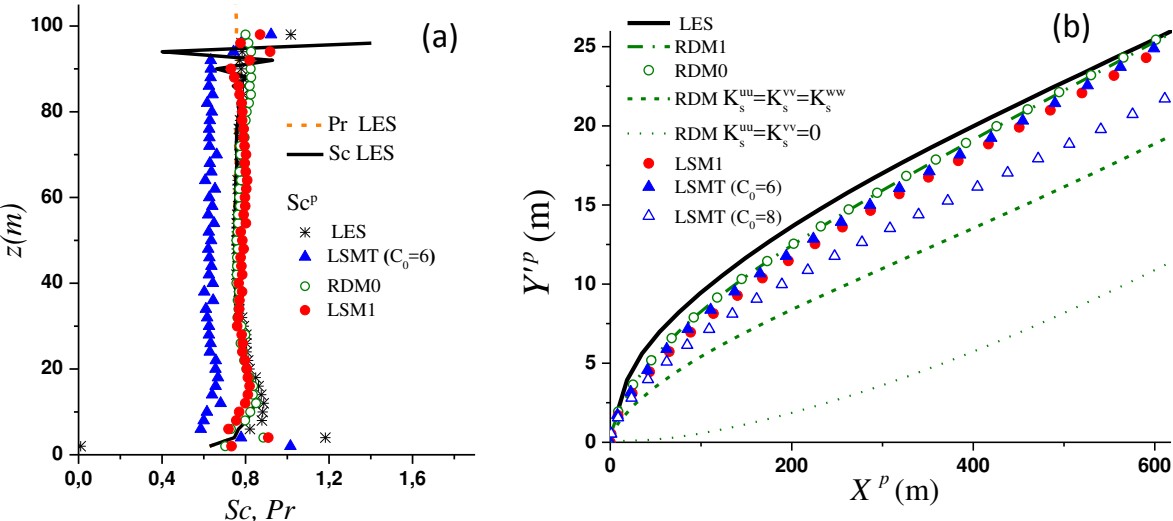

**Figure 12.** (a) Prandtl number $Pr$ (dashed line) and Schmidt number $Sc$ (solid line), computed using Eulerian scalars. Symbols - Schmidt numbers $Sc$, computed using the Lagrangian particles in LES, LSMs and RDMs. (b) RMS of the crosswind position of particle $Y'^p = \left\langle (y^p - Y^p)^2 \right\rangle^{1/2}$ depending on the mean longitudinal position $X^p = \langle x^p \rangle$. Dashed lines - RDM with $K_s^{uu} = K_s^{yy} = K_s^{ww}$ and one-dimensional RDM $K_s^{uu} = K_s^{vv} = 0$.

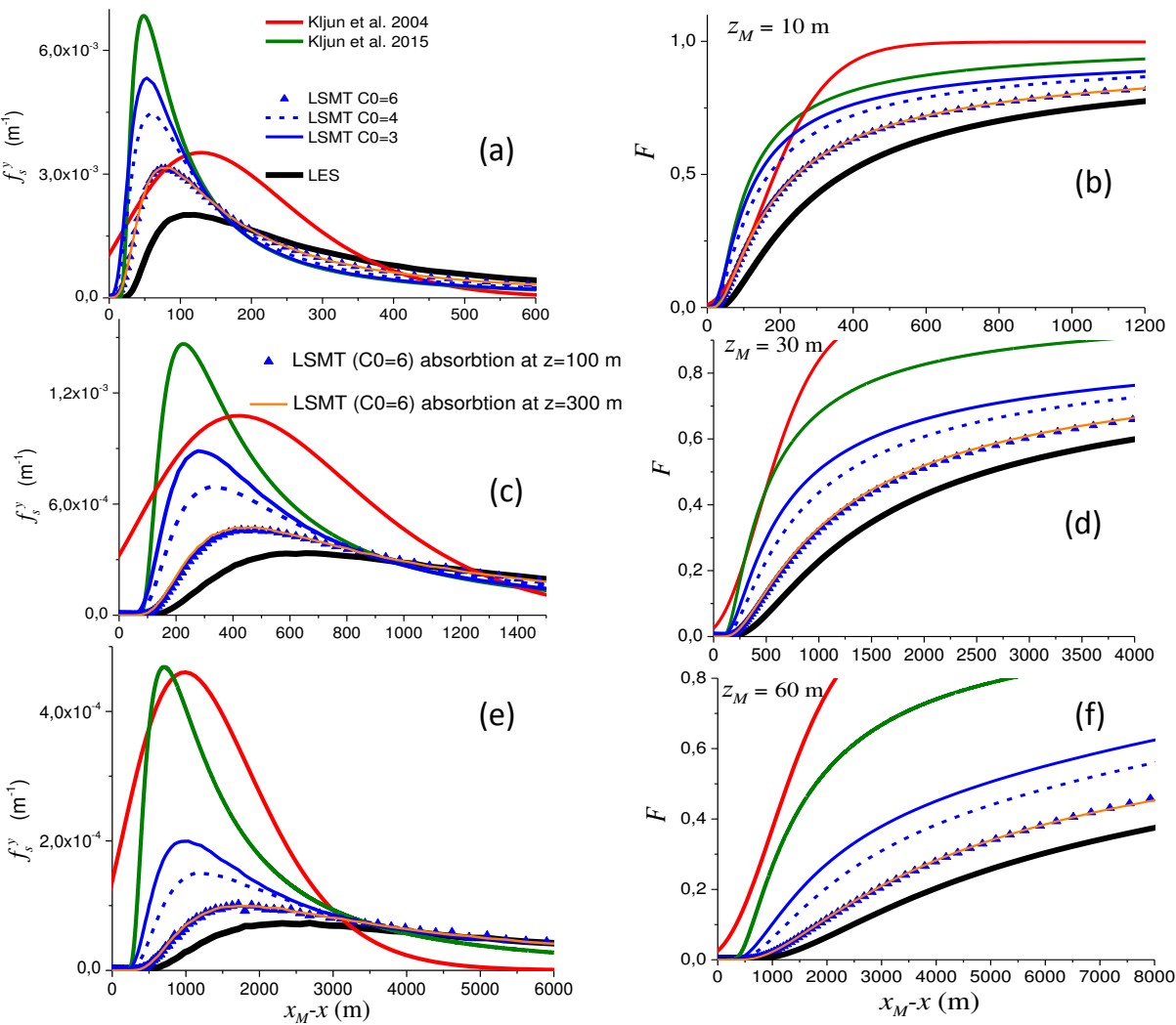

**Figure 13.** Crosswind-integrated footprints $f_s^y$ (a,c,e) and cumulative footprints $F$ (b,d,f) for sensor height $z_M = 10$ m (a,b), $z_M = 30$ m (c,d) and $z_M = 60$ m (e,f). Solid lines - LES with grid steps $\Delta_g$=2.0 m. Triangles - LSMT (Thomson (1987) model), $C_0 = 6$, absorbtion at z=100 m. Orange curves LSMT, $C_0 = 6$, absorbtion at z=300 m. Dashed blue lines - LSMT, $C_0 = 4$. Solid blue lines - LSMT, $C_0 = 3$. Red lines - parameterisation (Kljun et al., 2004). Green lines - parameterisation (Kljun et al., 2015).

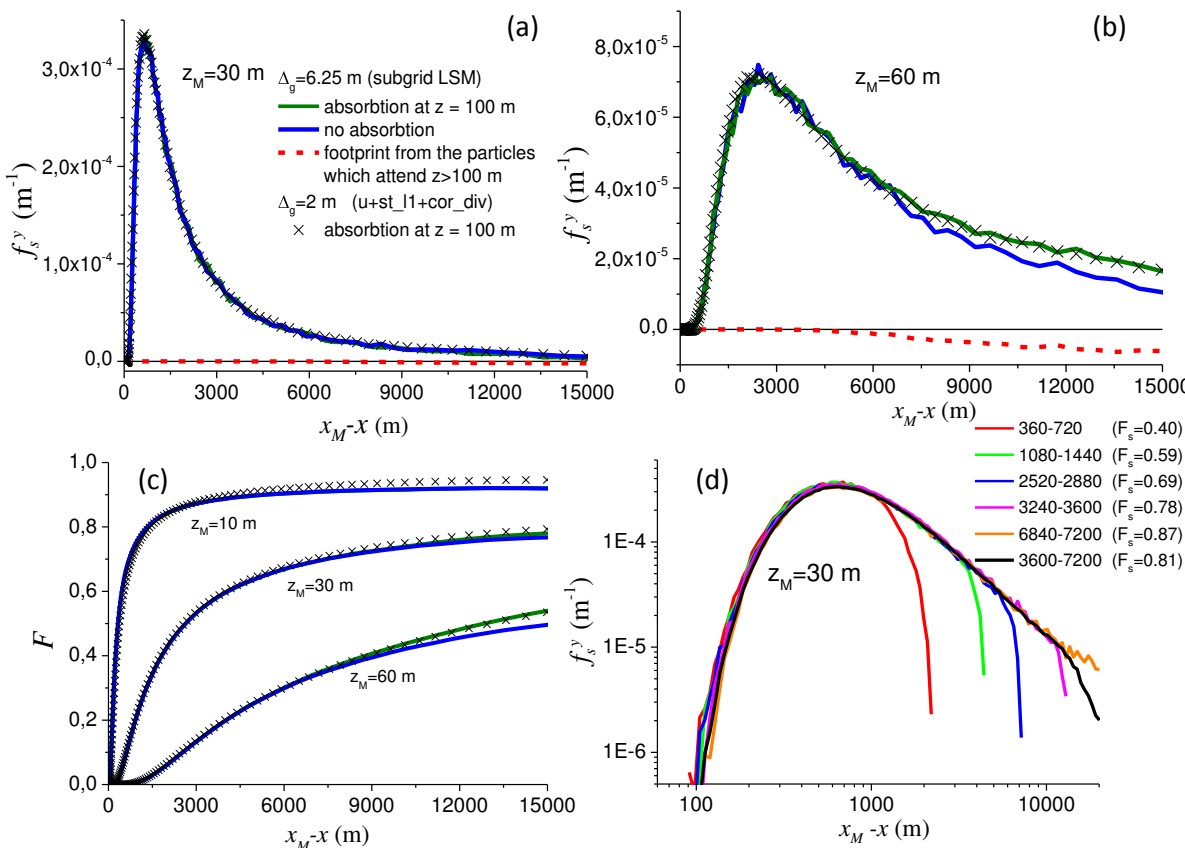

**Figure A1.** The footprint functions $f_s^y$ (a,b) and the cumulative footprints $F$ (c) obtained without the prescribed absorbtion (blue lines) in comparison with the results of simulation where the absorbtion is imposed at the level $z = 100$ m (green lines). Red dashed lines are the footprints from the particles which attained the level $z = 100$ m. (d) - Footprints obtained with the different intervals of averaging $[t_1, t_2]$ (shown in seconds in the legend), the normalized vertical concentration fluxes $\langle F_s(z_M)/F_s(0)\rangle_{[t_1,t_2]}$ are shown in brackets.