# Peer review of "Large-eddy simulation and stochastic modelling of Lagrangian particles for footprint determination in stable boundary layer"

_Geoscientific Model Development, 2016_

## Referee Comment (RC1) · Anonymous Referee #1 · 4 Apr 2016

The reviewed manuscript is generally well written, structured and provides all necessary details for the reproduction of the presented simulations.

I found just a few places where minor corrections would be desirable.

1) In Introduction, it is difficult to comprehend notations with regards to particles and model coordinates.

2) A new figure would help to explain the analysis and experiment set up. E.g. it is unclear why x_M is a vector but x,y not in Eq. (1). It is hard to understand what the coordinates of particles are and how the weight areas are computed. The figure should refer to Eq. (1) (2) and (3) and to the description on the page 4 lines 20-35.

[Figure]

3) Explain what "ensemble average" means in the context of the study.

4) Why is the index "p" used both as subscript and superscript in Eq. (7) and later on. Could you make notations more homogeneous?

5) Page 6, lines 2-3. The sentence is not quite clear. What will happen if a particle leaves the volume and then reappears again in the same volume during the unit time interval? Will it be counted as a new particle? Or do you mean something different under "appearing ... during unit time interval".

6) The sentence between Eqs. (8) and (9) is impossible to understand.

7) Page 9, line25. Use "provides better agreement" instead of "leads to better coincidence".

8) Section 4.2.3, also 5.1.4 and 5.1.5. It would be useful to place a discussion here into some experimental context referring to correlations between resolved and unresolved velocities (or velocities and stresses on different spatial scales) , e.g. the work by Charles Meneveau and co-authors (Meneveau and Katz in Annu. Rev. Fluid Mech. 2000. 32:1–32).

9) Use Figures instead of Pictures in the paper.

10) General remark in connection to Figure 6. Figure 6 shows a negative footprint. It is hard to understand the physical meaning of the negative values. Could you include a paragraph discussion this aspect?

11) in several places, e.g. line 24 at the page 11, the Equation number is referred without "Eq." so that it is difficult to understand what those numbers are for.

---

## Author Comment (AC1) · 13 Apr 2016

Please, see our response in the attached (supplement) file. Corrected version of manuscript is also included.

Please also note the supplement to this comment: http://www.geosci-model-dev-discuss.net/gmd-2016-34/gmd-2016-34-AC1-supplement.zip

---

## Referee Comment (RC2) · Anonymous Referee #1 · 17 Apr 2016

I'm satisfied with the answers to the RC#1 comments. The manuscript has gained in clarity and contextual connections to fluid mechanics. I have no further comments and want to see this important study published.

———————————————

---

## Referee Comment (RC3) · Anonymous Referee #2 · 20 Apr 2016

The manuscript by Glazunov et al. focuses on scalar dispersion and flux footprint functions in the case of the stable atmospheric boundary layer based on Large-Eddy Simulation (LES) and Lagrangian stochastic modelling. The approach is fairly straightforward and I see no scientific problems. I enjoyed reading the paper; in particular, the paper is well organized and well written, the presentation of the material is clear and concise. I am entirely in favor of publishing this paper in the Geosci. Model Dev. I only have a few minor comments, which authors should take into account.

Page 1, line 18. Replace 'the near-surface flux' by 'the surface flux' because it's defined for z=0 (cf. "L is the Obukhov length at the surface" on page 16, lines 1-2).

Page 1, lines 18-19. Replace 'denoting the ensemble averaging' by 'denote a

time/space average'.

Page 3, line 24. Although abbreviations 'LSM' and 'RDM' are defined in the abstract and later on page 7, they should be also introduced in the text on first occurrence.

Figures 1-10. I recommend use color version of the plots (similar to Fig. 11) instead black and white.

I believe that authors can clarify these points before finalizing the manuscript.

---

## Author Comment (AC2) · 20 Apr 2016

Response to Anonymous Reviewer #2

Authors are grateful to the referee for a high assessment of the article. Accordingly to the comments following changes were made:

**1)** *Page 1, line 18. Replace 'the near-surface flux' by 'the surface flux' because it's defined for z=0 (cf. "L is the Obukhov length at the surface" on page 16, lines 1-2).*

It was done.

**2)** *Page 1, lines 18-19. Replace 'denoting the ensemble averaging' by 'denote a time/space average'.*

Agree. Accordingly to comments of Reviewers #1 and #2 this paragraph was rewritten (see our response #1). In new version of manuscript we avoid ensemble averaging notation in Introduction.

**3)** *Page 3, line 24. Although abbreviations 'LSM' and 'RDM' are defined in the abstract and later on page 7, they should be also introduced in the text on first occurrence.*

In new version of the paper the abbreviations 'LSM' and 'RDM' are introduced on page 2, lines 12-14.

**4)** Figures 1-10. I recommend use color version of the plots (similar to Fig. 11) instead black and white.

Figures 2-6 and 8,10,11 (in previous version 1-5 and 7,9,10) were colorized. Figures 7 and 9 contain a few number of curves, so they remain to be black and white.

Additionally, the typo was corrected in Eq. (49). ($\xi_i^p$ was replaced to $\xi_3^p$)

Corrected version of the paper is attached (see, pdf file in supplement).

Please also note the supplement to this comment:
http://www.geosci-model-dev-discuss.net/gmd-2016-34/gmd-2016-34-AC2-supplement.pdf

[Figure]

[Figure]

**Supplement:**

[revised manuscript text omitted]

---

## Referee Comment (RC4) · Anonymous Referee #3 · 10 May 2016

**Geosci. Model Dev. Discuss.**

Manuscript:          gmd-2016-34
Title:               Large-eddy simulation and stochastic modelling of Lagrangian particles
                     for footprint determination in stable boundary layer

**Summary Evaluation**

Originality:                Good
Technical quality:          Good
Clarity of presentation:    Fair
Practical significance:     Good
Recommendation:             Major Revisions – I would like to review the revised manuscript

**General Comments**

In this paper, Large-Eddy Simulation (LES) is used in order to determine flux footprint functions in a stable boundary layer for three different sensor heights. One individual case has been investigated and from this quite general conclusions are drawn. Results are compared to those from existing footprint models based on Lagrangian stochastic particle dispersion modelling, LPDM (in different sophistication) and a scaling-based footprint parameterisation applicable for a wide range of stability and receptor heights. It is concluded that the recovery of small-scale (sub-filter scale) turbulent motions and hence the spatial scale is relevant for the quality of the LES results and that Kolmogorov's constant for the Lagrangian structure function in the inertial subrange ($C_0$) has a strong impact on the footprints from LPDMs.

As the LES is treated as 'truth' (reference), a much more substantial discussion of the LES quality must be provided (major comment 1). Given sufficient information on this matter, the LES results for footprints in stable conditions could be used as a basis for fitting footprint parameterisations. Such data are very much welcomed. However, for a statistically relevant data set, not only one single simulation, but several simulations for a broader range of stable conditions would be necessary. It would be great to see such data being made available. Also, it seems that some ad hoc assumptions (e.g., 'absorption condition', normalisation) are employed, which need further explanation and consideration (major comments 2 and 3). The results concerning Kolmogorov's constant for the structure function in the Inertial Subrange are probably somewhat overstated (major comment 4). Finally, when comparing to other models, it appears that the authors have not correctly reproduced one of these 'other models' (major comment 5).

Before the paper can be recommended for publication in GMD, these major comments and some minor comments must be addressed. It is also suggested that the language should be assessed by a native speaker.

**Main Comments**

*(1) Model validation and argumentation of approaches:*
All conclusions are based on the inherent assumption that the LES provides the 'truth' – even if quite a number of ad hoc assumptions are being made in order to produce the LES-based footprints. For example:

- corrections of advection speed due to subgrid-scale turbulence (Eq. 36) are applied only in the lowest grid layer [why exactly one?], p. 13, l. 16 – and somehow 'implemented' in the lowest three layers, p. 13, l. 20;
- furthermore, this correction is based on using a Langevin type of approach (Eq. 28), which employs a particular value for Kolmogorov's constant for the structure function in the Inertial Subrange [$C_0$] (which is not specified for this application);
- a further 'correction' is applied (Eqs. 33 and 34) with a somewhat arbitrary coefficient c=0.5, p. 14, l. 3
- particles are released at $z_0$ = 0.1 m, but reflected 'at the ground' (p. 12, l.17). Should this mean z = 0 m? And if so, how are the velocity statistics being evaluated below $z_0$?

All these settings are likely good (or reasonable) choices but should be substantiated. When serving as a reference for footprint calculations, the LES should be validated on a forward dispersion case from the literature. The reasoning for using LES as a reference comes from requiring 'scale invariance', i.e. independence of the results from grid resolution (p. 3, l. 17 – and Figs. 3c, d). This, first of all, is a good criterion in the absence of any true reference – but one would want to see to what degree the above choices influence this independence.

*(2) Absorption condition:*
Please provide more information on the absorption height and its impact (p. 12, l. 20 ff). I.e., provide a graph or a reference and list the tests undertaken confirming that "...the upper boundary condition does not have a large impact on the results of calculations of footprints...".
It seems that particles are absorbed at the absorption height and hence removed from the simulation domain. Figure 2 suggests that there is no vertical flux above the absorption height. However, turbulent fluxes should decline almost linearly from their surface value upwards to approach zero at the boundary layer height (i.e. in this case at z = 180 m and not at z = 100 m, cf. Stull, 1988 or Beare et al., 2006).

If the particles cannot reach the boundary layer height, they cannot be reflected at this height and cannot return to the surface. The consequence is that footprints consist of upwards flowing particles only. If so, this would result in an unrealistic increase in extent of flux footprints as downward flowing particles would weigh out upward flowing particles (when evaluating the vertical flux), with increasing tendency to do so with increasing distance from the measurement location. Please clarify how this is handled regarding the footprints from the LES.

*(3) Normalisation of footprints:*
On p. 13, l. 14, it is shown that the integral over the footprint function is normalised to one. Does this include negative footprint values, too? Or are these treated separately as mentioned on p. 15? Please clarify. The absolute values of the footprint function and hence the cumulated footprint will depend on how negative values are treated. Observed differences in the absolute footprint function values for different footprint approaches (cf. Fig. 11) may be partly due to differences in normalisation procedures.

Also, is there a threshold value for the distance from the measurement up to where footprint values are considered? The 'flat' trend of the cumulative footprint values suggests that the footprint function would only completely diminish at very large distances from the measurement location. If a threshold value is set, again the selected value will have an impact on the normalisation and the absolute value of the footprint function. Please provide more information on the applied procedure.

*(4) Kolmogorov's constant for the structure function in the Inertial Subrange [$C_0$]:*
First of all, this constant is referred to as 'Kolmogorov constant', a name that is usually associated with that in the energy spectra in the Inertial Subrange (and has a value of approximately 1.5). The authors discuss the range of proposed values in the literature, and it is felt that i) the paper by Rizza et al. (2010) might be a valuable addition to the discussion of possible values in the PBL and ii) the employed value in the LES subfilter correction (Eqs. 28 ff) should be provided. However, in the present paper it is demonstrated that the results of the LSMs (and in particular LSMT) are sensitive to the choice for $C_0$ – which is per se not particularly new (see, e.g., Rotach et al. (1996) who have sought the 'optimal' value based on comparison to water tank (dispersion) measurements of Willis and Deardorff – and many others, such as Du et al. (1995), Reynolds (1998), as cited in Rizza et al). If indeed the LES were fully validated and all the choices substantiated (see major comment 1), the present simulations would correspond to 'one more tessera' in the picture of a possible non-universality of $C_0$, be it due to stability dependence or employed time scales (outside those corresponding to the Inertial Subrange). It is felt that the conclusions drawn in the present paper (one 'case' – even with three heights) do not warrant the quite general conclusions drawn (p. 21, l. 18), i.e. 'the optimal value is found to be close to 6'.

*(5) Footprints plotted in Fig. 11:*
The footprints plotted in Fig. 11 of the manuscript and listed as Kljun et al. (2015) do not coincide with FFP model results. Plotted below are footprints derived from FFP for the input values mentioned in the manuscript, and optimised parameters for neutral and stable conditions as listed in Kljun et al. (2015). (Note: using the universal FFP parameters, e.g. from the online footprint tool still results in different footprints than those plotted in Fig. 11). It can be seen in Fig. R1 that the peak location of FFP fits very well the peak of LSMT with $C_0$=3 in Fig. 11. Footprint peak values, however, do differ, especially for larger measurement heights. Regarding the absolute values of these peak values please see major concerns (2 and 3) above. Also, the model of Kljun et al. (2004) is outdated; issues in stable conditions were known and were one of the reasons for the update to the model of Kljun et al. (2015).

As FFP compares well with the Lagrangian footprint model it is based on (see Fig. R2), and as different settings of $C_0$ produce similar shifts in footprints in LPDM-B (Kljun et al. 2002) and the LPDM used in this study (Fig. R2) – the main question boils down to: what is the 'ultimate truth' and what should a footprint parameterisation be based upon? (See comments above.) This is a very important question and I suggest that the authors highlight this fact even more in their manuscript.

[Figure]

Fig. R1: Crosswind-integrated flux footprints for the three measurement heights as calculated with the parameterisation FFP of Kljun et al. (2015). The footprints are calculated using the FFP code available online at footprint.kljun.net and with the neutral/stable set of parameters.

[Figure]

Fig. R2: Crosswind-integrated flux footprints for the three measurement heights as calculated with the parameterisation FFP (black line) of Kljun et al. (2015) and with LPDM-B of Kljun et al. (2002), setting $C_0 = 3$ (blue line) and $C_0 = 6$ (red line).

**Minor Comments**

(a)  The term "Analytical footprint model" is commonly used for footprint models based on analytical solutions of the diffusion equation by applying a K-theory approach. This is a distinctly different approach than used in the models of Kljun et al. (2004, 2015). The latter models are footprint parameterisations. Please correct throughout the manuscript.

(b)  p. 2, l. 5: '…commonly, the application of these models is limited by the constant flux approximation': this is not true at least for the Kljun et al. papers cited above.

(c)  p. 5, l. 26: If reference is made to 'the lake', this lake must be introduced beforehand. It is not appropriate to explain in brackets that the author apparently works on a 'lake problem'.

(d)  p. 8, l. 15: Euclidean: spelling?

(e)  According to Eq. 2, fys corresponds to the crosswind-integrated footprint. Please use this well-established term rather than 'crosswind averaged footprint' (e.g. p. 14, l. 3 or p. 20, l. 18). Further, in the captions of Figs. 9 and 11, the graphs are referred to as "One-dimensional footprints fys". This would suggest that the footprint at y=0 is plotted. Please clarify.

(f)  Wind profile: from Figure 1, it seems that simulated wind speeds in the surface layer part of the domain are smaller than the 'standard' wind profile for stable conditions (e.g., Stull, 1988; Högström, 1996). Please add a couple of sentences to explain why.

**References**

Beare, R.J.,MacVean,M.K., Holtslag, A.A.M., Cuxart, J., Esau, I., Golaz, J.C., Jimenez,M.A., Khairoutdinov, M., Kosovic, B., Lewellen, D., Lund, T.S., Lundquist, J.K., McCabe, A., Moene, A.F., Noh, Y., Raasch, S., Sullivan, P. : An Intercomparison of Large-Eddy Simulations of the Stable Boundary Layer. Boundary-Layer Meteorol., 118, 2, 247–272, 2006

Högström, U.: Review of Some Basic Characteristics of the Atmospheric Surface Layer, Bound.-Lay. Meteorol., 78, 215–246, 1996

Kljun, N., Rotach, M. W., and Schmid, H. P.: A 3D Backward Lagrangian Footprint Model for a Wide Range of Boundary Layer Stratifications, Bound.-Lay. Meteorol., 103, 205–226, 2002

Kljun, N., Rotach, M. W., and Calanca, P.: A Simple Parameterisation for Flux Footprint Predictions, Bound.-Lay. Meteorol., 112, 503–523, 2004

Kljun, N., P. Calanca, M.W. Rotach, H.P. Schmid: A simple two-dimensional parameterisation for Flux Footprint Prediction (FFP), Geosci Model Dev., 8, 3695-3713, 2015

Rizza et al.: Physica A, 2010. doi:10.1016/j.physa.2010.05.059

Rotach, M. W., Gryning, S.-E., and Tassone, C.: A Two-Dimensional Lagrangian Stochastic Dispersion Model for Daytime Conditions, Q. J. Roy. Meteorol. Soc., 122, 367–389, 1996

Stull, R. B.: An Introduction to Boundary Layer Meteorology, Kluwer Academic Publishers, Dordrecht, the Netherlands, 680 pp., 1988

---

## Author Comment (AC4) · 1 Jun 2016

We are very grateful to reviewer for insightful analysis of our paper. All the comments are very professional and helped us to improve substantially the manuscript. The authors believe, that the material presented in the manuscript became better justified after the revision. Please, see our response in the attached (supplement) file. Corrected version of manuscript is also included.

Please also note the supplement to this comment:
http://www.geosci-model-dev-discuss.net/gmd-2016-34/gmd-2016-34-AC4-supplement.zip

---

## Author Response (AR2)

**Response to Anonymous Reviewer #1**

The authors are grateful to the anonymous reviewer for carefully reading the manuscript and the proposed corrections. Accordingly to the comments following changes were made:

**1) Reviewer:** *In Introduction, it is difficult to comprehend notations with regards to particles and model coordinates.*

**Answer:** First paragraph in Introduction (beginning from the line 17, page 1 up to the line 8, page 2) was reformulated as follows:

The relationship between the near-surface flux $F_s(x, y, 0)$ and the flux $F_s(x_M, y_M, z_M)$, measured in point $\mathbf{x}_M = (x_M, y_M, z_M)$, can be formalized via the footprint function $f_s$:

$$F_s(x_M, y_M, z_M) = \int_{-\infty}^{\infty} \int_{-\infty}^{\infty} f_s(x, y, x_M, y_M, z_M) F_s(x, y, 0) dx dy. \tag{1}$$

Traditionally, footprint functions $f_s^d(x^d, y^d, \mathbf{x}_M) = f_s(x, y, \mathbf{x}_M)$ are expressed in local coordinate system with the origin which coincides with the sensor position (here, $x^d = x_M - x$ is the positive upwind distance from the sensor and $y^d = y_M - y$ is the cross-wind distance, see Fig. 1a). In horizontally homogenous case these functions do not depend on $x_M$ and $y_M$. In ABL the surface area contributing to the flux is elongated in wind direction, therefore the cross-wind integrated footprint function $f_s^y$ defined as

$$f_s^y(x^d, z_M) = \int_{-\infty}^{\infty} f_s^d(x^d, y^d, z_M) dy^d, \tag{2}$$

is one of the most required characteristics for the practical use.

The measurements of the scalar flux footprint functions in natural environment are restricted (e.g., Finn et al.,1996;Leclerc et al., 1997, 2003; Nicolini et al., 2015) due to the necessity to conduct the emission and detection of artificial tracers. Besides, such measurements are not available for the stably stratified ABL where the area of the surface influencing the point of measurements increases.

Here we avoided introducing of the averaging notations, which bring no additional sense in original version of manuscript. Systems of coordinates are illustrated in additional schematic Fig. 1a.

**2) Reviewer:** *A new figure would help to explain the analysis and experiment set up.*

*E.g. it is unclear why $\mathbf{x}_M$ is a vector but x,y not in Eq. (1). It is hard to understand what the coordinates of particles are and how the weight areas are computed. The figure should refer to Eq. (1) (2) and (3) and to the description on the page 4 lines 20-35..*

**Answer:** New schematic figure was added (Fig.1). It was supplemented with the appropriate description of footprint evaluation algorithm in Section 2.1:

Schematic representation of the algorithm for the footprint function determination in LES is shown in Fig. 1. In accordance with Eq. (3) and the description above, the particle crossing the test area $\delta_M$ brings the impact into the value $f_s(x_S, y_S, \mathbf{x}_M)$ then the beginning of its modified trajectory shifted in a such way to superpose the point $\mathbf{x}_1^p$ with sensor position $\mathbf{x}_M$ belongs to the test area $\delta_S$. For example (see, Fig. 1b), red particle is counted while evaluation of the footprint value in point $(x_S, y_S)$, but blue particle is not counted. Such algorithm of averaging was selected because it permits to refine the footprint resolution in the vicinity of sensor independently on the area of $\delta_M$ using the assumption of some spatial homogeneity.

Besides, we added description of the grid used for footprint accumulation (see last paragraph of Sect. 2.1 in modified version of manuscript):

Nonuniform Cartesian grid $\mathbf{x}_{ij}^d = (x_i^d, y_j^d)$ (where, $-20 \leq i \leq 160$; $-120 \leq j \leq 120$), stretched with the distance from the sensor position, was selected for the footprint functions accumulation in the following sections of this paper. Grid was prescribed as: $(x_0^d, y_0^d) = (0,0)$; $x_i^d = \Delta_{x0}\gamma_x^{|i|} i/|i|$ and $y_i^d = \Delta_{y0}\gamma_y^{|j|} j/|j|$ if $i \neq 0$ and $j \neq 0$; $\Delta_{x0} = \Delta_{y0} = 2$ m; $\gamma_x = \gamma_y = 1.05$. This grid is independent on the LES model resolution and coincides with the footprint grids selected for all runs with LSMs and RDMs.

**3) Reviewer:** *Explain what "ensemble average" means in the context of the study.*

**Answer:** The following clarification was included after the Eq. (8) in Sect. 2.3:

Here $\left\langle u_i^{(p)} \right\rangle$ is the ensemble averaged Eulerian velocity at point $\mathbf{x}^p$. Note, that LSMs are assumed to be also applicable under the temporal evolution of turbulence statistics. In this paper we shall consider ABL as it approaches a quasi-steady state. Therefore, due to assumption of ergodicity, ensemble averaging can be replaced by averaging in time and in the directions of spatial homogeneity: $\langle \varphi \rangle \approx \langle \varphi \rangle_{x,y,t}$.

**4) Reviewer:** *Why is the index "p" used both as subscript and superscript in Eq. (7) and later on. Could you make notations more homogeneous?*

**Answer:** The notation $s_p$ was used to denote evaluated value of scalar concentration by the number of particles (subscript $p$ was not connected with the superscript $p$). In new version of manuscript we denote this value $s_P$ (with capital $P$) to avoid misunderstanding.

**5) Reviewer:** *Page 6, lines 2-3. The sentence is not quite clear. What will happen if a particle leaves the volume and then reappears again in the same volume during the unit time interval? Will it be counted as a new particle? Or do you mean something different under "appearing ... during unit time interval".*

**Answer:** The word "appearing" was replaced by word "ejected". Here we mean the new particles which were added during the run (appearing of new particles imitates the external source of scalar concentration).

**6) Reviewer:** *The sentence between Eqs. (8) and (9) is impossible to understand.*

**Answer:** This sentence was rewritten as follows:

Single particle first-order LSM is formulated as follows. Velocity $u_i'^p$ is described by the stochastic differential equation: ...

**7) Reviewer:** *Page 9, line 25. Use "provides better agreement" instead of "leads to better coincidence".*

**Answer:** It was done.

**8) Reviewer:** *Section 4.2.3, also 5.1.4 and 5.1.5. It would be useful to place a discussion here into some experimental context referring to correlations between resolved and*

*unresolved velocities (or velocities and stresses on different spatial scales) , e.g. the work by Charles Meneveau and co-authors (Meneveau and Katz in Annu. Rev. Fluid Mech. 2000. 32:1–32).*

**Answer:** The paper (Meneveau and Katz, 2000) is devoted to the a priory testing of the scale-similarity approaches for subgrid modelling of Eulerian dynamics in LES. The reference to this paper is useful in Section 3.1 where the mixed subgrid/subfilter model is introduced. So, the following text was added after the Eq. (17):

The a priori tests using the data of laboratory measurements show that scale- similarity models with Gaussian or box filters provide correlation typically as high as 80% between real and modeled stresses (see, overview in Meneveau and Katz, 2000). The significant part of this correlation can be attributed to non-ideality of the spatial filter and use of common information for computing both the real and modeled stresses (see, Liu et al., 1994). The discrete spatial filter used in this study has a smooth transfer function in spectral space, so it can be supposed that the scale-similarity part of Eq. (18) is mainly responsible for the influence of velocity fluctuations belonging to "subfilter" scales.

In Section 4.2.3 we introduced next clarification of high correlation between subfilter velocity and resolved velocity:

In the previous subsection the recovered "subfilter" part of velocity $\mathbf{u}'' = \mathbf{u}^* - \overline{\mathbf{u}}$ and so the subfilter Lagrangian velocity $\mathbf{u}''^{(p)}$ were highly correlated with the resolved velocity $\overline{\mathbf{u}}$ in time and space. This is due to the specifics of spatial filter (Eq. 24) used for the recovering given by Eqs. (25, 26). This filter has a smooth transfer function in spectral space. The analogous effects of non-ideal filters in LES which lead to the high correlations between modelled and measured turbulent stresses were obtained and discussed earlier in (Liu et al., 1994) and (Meneveau and Katz, 2000), where the laboratory data of turbulent flows were studied.

Section 5.1.4 contains the description of the universal function used for correction of dissipation profile. New reference was added to the paper which contains the measurement data of similar nondimensional function. The text after Eq. (43) was modified as follows:

Previous LES studies of stable ABL (e.g., Beare et al., 2006) also give neglectfully small values of the transport terms in TKE balance. The experimental confirmation of the validity of Eq. (42) can be found in (Grachev et al., 2015), where the dissipation in stable ABL was estimated using the spectral analysis of longitudinal velocity in inertial range. In accordance with this paper: $\tilde{\epsilon} \approx \phi_m$, that is almost indistinguishable from Eq. (42) within the accuracy of the experimental data and the ambiguity of the method of dissipation evaluation.

Section 5.1.5 contains the results of the evaluation of diffusion coefficients. Here these coefficients are presented as dimensional values and are specific for the modelled flow. There are no available experimental data on the values of horizontal diffusivity in horizontally homogeneous stable ABL. At least, authors are not aware of such measurements.

**9) Reviewer:** *Use Figures instead of Pictures in the paper.*

**Answer:** It was corrected.

**10) Reviewer:** *General remark in connection to Figure 6. Figure 6 shows a negative footprint. It is hard to understand the physical meaning of the negative values. Could you include a paragraph discussion this aspect?*

**Answer:** Next explanation was included into first paragraph of Section 4.3:

The negative values of scalar flux footprint show what the vertical turbulent transport of the scalar emitted in the relevant area is basically directed from the upper levels down to the surface. For example the positive surface concentration flux in this area will lead to negative anomaly of the turbulent flux measured in the sensor position. This does not contradict the diffusion approximation of the turbulent mixing, because mean crosswind advection at the upper levels can produce the positive vertical concentration gradient to the right of near-surface wind.

**11) Reviewer:** *in several places, e.g. line 24 at the page 11, the Equation number is referred without "Eq." so that it is difficult to understand what those numbers are for.*

It was corrected.

Additional references were included into bibliography:

Grachev, A. A., Andreas, E. L., Fairall, C. W., Guest, P. S. and Persson, P. O. G.: Similarity theory based on the Dougherty–Ozmidov length scale. Q.J.R. Meteorol. Soc., 141, 1845–1856, 2015.

Liu S., Meneveau C., Katz J.: On the properties of similarity subgrid-scale models as deduced from measurements in a turbulent jet. J. Fluid Mech. 275, 83–119, 1994.

Meneveau C. and Katz J.: Scale-invariance and turbulence models for large-eddy simulation. Annu. Rev. Fluid Mech., 32, 1–32, 2000. Michalek W.R., J.G. M.Kuerten, J.C.H. Zeegers, R.Liew, J.Pozorski, B.J. Geurts: A hybridstochastic-deconvolution model for large-eddy simulation of particle-laden flow. Physics of Fluids, 25, 123302, 2013

Response to Anonymous Reviewer #2

Authors are grateful to the referee for a high assessment of the article. Accordingly to the comments following changes were made:

**1) Reviewer:** *Page 1, line 18. Replace 'the near-surface flux' by 'the surface flux' because it's defined for z=0 (cf. "L is the Obukhov length at the surface" on page 16, lines 1-2).*

**Answer:** It was done.

**2) Reviewer:** *Page 1, lines 18-19. Replace 'denoting the ensemble averaging' by 'denote a time/space average'.*

**Answer:** Agree. Accordingly to comments of Reviewers #1 and #2 this paragraph was rewritten (see our response #1). In new version of manuscript we avoid ensemble averaging notation in Introduction.

**3) Reviewer:** *Page 3, line 24. Although abbreviations 'LSM' and 'RDM' are defined*

*in the abstract and later on page 7, they should be also introduced in the text on first oc-currence.*

**Answer:** In new version of the paper the abbreviations 'LSM' and 'RDM' are introduced on page 2, line 12-13.

**4) Reviewer:** Figures 1-10. I recommend use color version of the plots (similar to Fig. 11) instead black and white.

**Answer:** Figures 2-6 and 8,10,11 (in previous version 1-5 and 7,9,10) were colorized. Figures 7 and 9 contain a few number of curves, so they remain to be black and white.

Additionally, the typo was corrected in Eq. (49). ($\xi_i^p$ was replaced to $\xi_3^p$)

Corrected version of the paper is attached (see, pdf file in supplement).

**Response to Anonymous Reviewer #3**

We are very grateful to reviewer for insightful analysis of our paper. All the comments are very professional and helped us to improve substantially the manuscript. The authors believe, that the material presented in the manuscript became better justified after the revision.

We would like to address two topics raised by the reviewer in a different order than presented in the review. These comments concern the validity of the simulation results (minor comment f) and the correctness of the presented results in Figure 11. It will be impossible to proceed with other comments until we consider these issues.

1) **Reviewer: (f) Wind profile: from Figure 1, it seems that simulated wind speeds in the surface layer part of the domain are smaller than the 'standard' wind profile for stable conditions (e.g., Stull, 1988; Högström, 1996). Please add a couple of sentences to explain why.**

**Answer:** This comment is suggested as minor but from the authors point of view it is of major importance. There is no sense in discussing anything else, if the numerical model used in this study produces wrong results. We have added the data from eight different LES models in Fig.2 (Fig.AR1 here and Fig.1 in the original version of the manuscript). These data were obtained during LES intercomparison experiment GABLS-1 and are available online at:

http://gabls.metoffice.com/lem_data.html. We used the data for 3.125 m resolution, because they were presented for the largest set of the models and because the results do not change substantially with the following grids refining. Wind velocities from the other models shown in Fig.2 (Fig.AR1) are rotated 35 degrees clockwise in accordance with the setup of our runs. The results from the LES model used for the current study fit with the results of the other models very well. Besides, our model gives a good scale invariance, which is not the case for some models presented at http://gabls.metoffice.com/means_125.html. Mean wind profile computed in accordance with [Högström, 1996] is shown by the vertical dashes in Fig.AR1, this profile almost coincides with the longitudinal velocity obtained in LES. Accordingly, the authors have no reason to doubt the results of the simulations. In the opposite case it would have been questionable the LES methodology for the stable boundary layer investigation.

**Corrections:**

Figure 2 (AR1, former Fig.1) was modified by adding the data from other LES models referred

to above.

Following clarification was inserted into the text (p.13 l.21 - p.14 l.3):

This setup is based on the observation data (see, [Kosoviĉ and Curry, 2000]). As it was shown in [Beare et al., 2006], the LES results obtained under the same conditions with the different models converged with the higher grid resolutions. Later, this case was used for testing the LES models e.g. in [Maronga et al., 2015, Zhou and Chow, 2012, Bhaganagar and Debnath, 2015] and many others and for the improvement of subgrid modelling e.g. in [Basu and Porté-Agel, 2006, Zhou and Chow, 2011, Kitamura, 2010]. The LES model presented here was tested earlier under the non-modified setup of GABLS in [Glazunov, 2014], where the turbulent statistics above a flat surface and above an urban-like surface were investigated. In all of these studies, LES results were in agreement with the known similarity relationships for the stable ABL. This allows to consider the LES data for GABLS as a reference case for testing of the approaches utilizing the statistical averaging of the turbulence (e.g., see [Cuxart et al., 2006], where the intercomparison of single-column models was performed). Several of nondimensional relationships in stable ABL were collected and presented in [Zilitinkevich et al., 2013]. Considered case is also included in the LES database for this study and fits well with the different stability regimes after the appropriate normalization. Therefore, the results obtained in this particular case can be generalized for many cases due to similarity of the stable ABLs. Besides, the presented simulations are easily reproducible and they can be repeated using any LES model which contains the Lagrangian particle transport routines.

The mean wind velocity and the potential temperature, calculated with the different spatial steps $\Delta_g$, are shown in Fig. 2 The model slightly overestimates the height of the boundary layer at coarse grids, however, the wind velocity near the surface is approximately the same in all runs. As one can see from the Fig. 2, the results of simulation are in good agreement with the results from other LES presented in [Beare et al., 2006] (see, http://gabls.metoffice.com for more information). Mean wind profile computed in accordance with [Högström, 1996] is shown in Fig. 2 by the vertical dashes, in the surface layer part of the domain this "standard" profile for the stable conditions almost coincides with the longitudinal velocity obtained in LES.

2) **Reviewer: Finally, when comparing to other models, it appears that the authors have not correctly reproduced one of these 'other models' (major comment 5)**.

**Answer:** Most likely there was an unfortunate misunderstanding. In our paper Obukhov length $L$ was defined as:

$$L = -\frac{U_*^3 \Theta_0}{g Q_s},$$

where $Q_s$ is the kinematic potential temperature flux at the surface, $g$ is the acceleration of gravity and $\Theta_0$ is the reference potential temperature. This definition does not include von Karman constant $\kappa \approx 0.4$ in the denominator. It was pointed out in the original version of the paper, see page 16, line 2:  '...note, that the von Karman constant is not included in the definition of the length $L$ here and later...' and in the definition of the local Obukhov length Eq. (40). Such definition of the Obukhov length scale is used alternatively (see, e.g. [Zilitinkevich et al., 2013] Eq.(41)) to its convenience when operating with the stably stratified flows outside the surface layer.

In the original version of the paper we wrote (p.20 l.15–17): "Nevertheless, the final approximations [Kljun et al., 2004] and [Kljun et al., 2015] contain the input parameters, which can be determined from LES: the boundary layer height $z_i \approx 180$ m, Obukhov length $L/\kappa \approx 120$ m, friction velocity $U_* \approx 0.27$ m/s and roughness parameter $z_0 = 0.1$ m. These values were substituted into parameterisations [Kljun et al., 2004] and [Kljun et al., 2015]".

Here $L$ is defined without $\kappa$ in denominator and the number 120 is the appropriate value for FPP, where this constant is included (see, [Kljun et al., 2015], Appendix B).

We performed the calculations of the footprint functions again using the online tool
http://footprint.kljun.net/ffp.php
and got nearly the same results, as were presented previously in Fig.11 (see short dashed lines in Fig.AR2). The next parameters were used:
L = 120
u_star = 0.272
sigma_v = 0.44
z0 = 0.1
u_mean = 0
h = 180
After substituting L= 120×0.4 =48 m into FPP calculator we got the results shown in Fig.AR2 by the dot-dashed lines. These results are very close to those presented in the review. One can find the values of the Obukhov length which is characteristic for the simulated case provided by other LES models at http://gabls.metoffice.com/times_200.html . It is also close to 120 m (48

m, as defined in our paper).

According to the analysis above we believe that the model FPP of [Kljun et al., 2015] was applied correctly in our paper. The Fig. 11 (Fig. 13 in the revised version) will remain unchanged.

**Corrections:** Nevertheless, to avoid misunderstanding we insert new equation (40) in Sect. 5.1.1 (the expression for $L$ provided above) and define this length scale explicitly with the following commenting:

Note, that the von Karman constant is not included in the definition of the length $L$ here and later (this alternative definition of the Obukhov length is often used along with the traditional one, see e.g. [Zilitinkevich et al., 2013] Eq.(41)).

*Starting from here, we shall follow the order of comments provided in the Review.*

**Major comments**

**(1) Model validation and argumentation of approaches**

**Reviewer: corrections of advection speed due to subgrid-scale turbulence (Eq. 36) are applied only in the lowest grid layer [why exactly one?], p. 13, l. 16 - and somehow 'implemented' in the lowest three layers, p. 13, l. 20**

**Answer:** LSM was implemented in the lowest three layers, but the additional stochastic component of velocity produced by this model was taken into account inside the first layer only during the computation of particle position. Only one layer because the aim of this procedure is to minimize use of computational resources without loss of quality and to test the validity of the Lagrangian particles transport with the minimal use of the nondeterministic terms. The presented results show that it is enough.

**Corrections:** In new variant of the paper we insert clarification and define this procedure explicitly (see, p. 15, l. 14-17 in new version of the paper):
For the curves marked "st_1l", the resultant velocity of the particles near the surface was calculated

as follows:

$$\vec{u}^p = \vec{u}^{(p)} + r(z^p)\vec{u}''^p,$$

where the function $r(z^p)$ is defined as $r(z^p) = (1 - z^p/\Delta_g)$ if $z_p < \Delta_g$, $r(z^p) = 0$ if $z_p \geq \Delta_g$ and $\vec{u}''^p$ is the random velocity component, calculated using the stochastic subgrid model.

**Reviewer: furthermore, this correction is based on using a Langevin type of approach (Eq. 28), which employs a particular value for Kolmogorov's constant for the structure function in the Inertial Subrange [$C_0$] (which is not specified for this application)**

**Answer:** We agree.

**Corrections:** Next sentence was included after the Eg.(28) l. 10-11 p. 11:
The parameter $C_0$ was specified to be equal to 6, because the stochastic part of the model (Eq. 28) is mainly responsible for spatial and time scales in an isotropic inertial subrange of the turbulence.

**Reviewer: a further 'correction' is applied (Eqs. 33 and 34) with a somewhat arbitrary coefficient c=0.5, p. 14, l. 3**

**Answer:** We agree that this coefficient was selected quite arbitrarily. Justification for this choice is that,

i) The results of footprint calculation are not very sensitive to this coefficient, see Fig. 4, where the crosses are the footprints, computed without correction ($c=0$). These footprint functions approximately coincide with the results of other methods applied. The main reasons for the use of correction in addition to the velocity recovering were discussed in Sect 4.2.2 (Spatial variability of scalar concentration inferred by Eulerian and Lagrangian methods).

ii) Other Lagrangian subgrid models (LSM implemented in the whole domain and RDM added to the new version of the paper) give similar results.

**Reviewer: particles are released at z0 = 0.1 m, but reflected 'at the ground' (p. 12, l.17). Should this mean z = 0 m? And if so, how are the velocity statistics being evaluated below z0?**

**Answer:** There are no physical arguments for a rigorous specifying of the values of turbulent statistics below $z_0$, because the roughness length is not more than the parameter in logarithmic velocity profile. In the model the values of turbulent subgrid energy and the dissipation of subgrid

TKE were interpolated linearly for $z > \Delta/2$ and were fixed at the values $\epsilon = \epsilon(\Delta/2)$ $E = E(\Delta/2)$ below. Our experience with LSMs and LES models shows that the details of fast mixing near the ground do not influence the footprints considered here, especially if the grid steps are small enough. For example, in the runs with $\Delta_g = 3.125$ and $2.0$ m one can substitute the vertical velocity inside the first layer by the value of reconstructed velocity $w^*$ at the level $z = \Delta_g$ and to perform all simulation without the stochastic terms at all, and when doing so it will lead to extremely overestimated mixing inside the first layer, but the footprints with $z_M = 10$ m and $z_M = 30$ m will be almost unchanged (not included in the paper, we can supply the appropriate figures or data if it is necessary).

**Corrections:** In the revised paper the procedure of the interpolation of the turbulent statistics inside the first layer is described explicitly (l. 28-29 p. 11) with the following commenting:

This procedure is rather arbitrary, but it does not have large impact on the results due to the small decorrelation time $T_L(\Delta_g/2)$. Besides, there are no physically grounded reasons for the justification of such interpolations in LES because the resolved velocity in the vicinity of surface is greatly corrupted by the approximation errors. Such procedures should be considered as an adjustments depending on the numerical scheme and on the subgrid closure.

**Reviewer: All these settings are likely good (or reasonable) choices but should be substantiated. When serving as a reference for footprint calculations, the LES should be validated on a forward dispersion case from the literature.**

**Answer:** Some examples of such validation were already included in the Supplements to this paper including the simulation results at very rough grids (see, the Supplement S1). It was commented in the Introduction. In this supplement Fig.S1.1(AR9) shows the results of the validation our LES model using [Willis and Deardorff, 1976] data. Figure S1.3(AR10) shows the simulated footprint function and the measured one in convective ABL (case (b) from [Leclerc et al., 1997]). To compare, one can find the results of resolution sensitivity tests with other LES model with embedded particles under the same conditions in ([Steinfeld et al., 2008], Fig.4). There are no available data on footprints in the stable boundary layer which is considered here.

**Reviewer: The reasoning for using LES as a reference comes from requiring 'scale invariance', i.e. independence of the results from grid resolution (p. 3, l. 17 - and Figs. 3c, d). This, first of all, is a good criterion in the absence of any true reference - but**

**one would want to see to what degree the above choices influence this independence.**

**Answer:** Although the authors consider the independence of the results from the grid steps to be the sufficient reason for the justification of the methods applied, nevertheless some imaginable chance exists, that all of these methods provide the 'scale invariance' but at the same time prevent the convergence to the true result. There is no possibility for the grounded rejection of this chance when the models are not free from the adjustable procedures. Subgrid LSM and LSM in the vicinity of the surface combined with our approach are not the exceptions.

Accordingly, we decided to investigate one more subgrid model (RDM), which is rigorously determined by the parameterisations which are already included into the Eulerian LES equations and do not contain any adjustable procedures or parameters. We obtained the results, which are in a close agreement with the results obtained before, except for some details inherent to RDM and known previously from the literature (see, Fig.AR7). Agreement of the different approaches (subgrid LSM, subgris RDM and the recovery of small-scale (sub-filter scale) turbulent motions) provides additional support for correctness of the results.

**Corrections:** New Section 4.2.4 'LES with subgrid RDM and the comparison of different approaches' (page 17, and the description of this model, page 11) was added to the paper. New Figure 7 (Fig.AR7) was included.

**(2) Absorption condition:.**

**Reviewer: Please provide more information on the absorption height and its impact (p. 12, l. 20 ff). I.e., provide a graph or a reference and list the tests undertaken confirming that "...the upper boundary condition does not have a large impact on the results of calculations of footprints...".**

**Answer:** The confirmation of this sentence was provided in the original version of the paper for the Lagrangian stochastic model (LSMT). See, orange curve in Fig. 11. Here, the absorption was applied above the boundary layer height (a very small portion of the particles can reach this height because there is no turbulent mixing above z =180 m).

Additional confirmation of the validity of our assumption can be done by analyzing the particles trajectories in LES.

In additional run we did not perform any removing of the particles during calculations. We evaluated separately the footprints from the particles trajectories that at least once reached the height z= 100 m. (see, Fig.AR8). As one can see, the footprint for $z_M = 10$ m is not influenced by the artificial boundary condition. The impact of the returned particles into footprint for $z_M = 30$ m is also very small. For the higher level ($z_M = 60$ m) the influence of the upper boundary condition is visible for the distances $x - x_M$ larger then 6 km. The positions and the values of the footprint peaks are not affected by the influence of the top boundary condition and are not directly connected to the value of the vertical turbulent flux at the appropriate levels. To confirm the last conclusion, we present a series of footprints computed with different intervals of averaging in time (see, Fig.AR8). Here, time (in seconds) from the beginning of the particles ejection is shown in the legend. The footprints are developed sequentially - the processes with the small time scales form the peaks first. The value of the vertical concentration flux normalized by its surface value is shown in brackets. One can see that while the total vertical flux grows approximately twice, the positions and the values of the peaks of the crosswind-integrated footprints remain unchanged.

**Corrections:** We added new Appendix A1 which contains the results of the test presented above.

To be more rigorous, the following sentence in the paper:

'It was verified previously that the upper boundary condition does not have a large impact on the results of calculations of footprints for the heights $z_M$ up to 60 m'.

was replaced by the following one:

'It was verified previously that the upper boundary condition does not have a large impact on the results of calculations of footprints for the heights $z_M$ up to 60 m and for the distances $x - x_M$, considered in this paper (see Appendix A1 and the test with LSM shown by the orange curves in Fig.13)'.

**Reviewer: It seems that particles are absorbed at the absorption height and hence removed from the simulation domain.**

**Answer:** Yes, it is really so.

**Reviewer: Figure 2 suggests that there is no vertical flux above the absorption height. However, turbulent fluxes should decline almost linearly from their surface value upwards to approach zero at the boundary layer height (i.e. in this case at z = 180 m and not at z = 100 m, cf. Stull, 1988 or Beare et al., 2006).**

**Answer:** The suggestion that the concentration flux declines linearly from their surface value upwards to approach zero at z = 100 m is nonrealistic under the conditions considered here because in this case all the particles will retained in the simulation domain in spite of the absorption. At large simulation times, more realistic final state will be the constant concentration flux from $z = z_0$ up to $z = 100$ m and zero flux if $z < z_0$. We say 'more realistic' because this assumption is based on another assumption that the concentration will reach some limit inside the layer 0–100 m. We did not obtain this state in the presented calculations ( the particles simulation time 2 h is not long enough) . The expected flux profile and the concentration profile in the considered case are beyond the scope of this paper, although it is a very interesting task which could be considered in the scope of similarity theory. Taking into account the local nature of the stably stratified turbulence, the authors do not exclude the possibility of nearly the linear profiles of the fluxes, as it is shown in modified Fig.3 (AR11). One can see, what the values of the flux in our simulation are very close to those predicted by Stull, 1988 or Beare et al. 2006 up to the heights $z \approx 60$ m.

In any case, significant differences between the "true" and the modelled footprints will be expected at very large distances from the measurements location, and the flux profile does not affect footprint close to the measurement point position (see, new Appendix A1 and Fig.AR8d).

**Reviewer: If the particles cannot reach the boundary layer height, they cannot be reflected at this height and cannot return to the surface. The consequence is that footprints consist of upwards flowing particles only.**

**Answer:** The footprints consist of the upward and the downward flowing particles except those, which already reached the specified level z=100 m. Due to the local nature of the stably-stratified turbulence, and due to the large vertical velocity gradient, the particles, which reach the level z=100 m will return back after a rather big time interval and in a very outlying position (see, new Appendix A1).

**Reviewer: If so, this would result in an unrealistic increase in extent of flux footprints as downward flowing particles would weigh out upward flowing particles (when evaluating the vertical flux), with increasing tendency to do so with increasing distance from the measurement location.**

**Answer:** Yes it is indeed so, but for the levels and for the distances which are not considered here. (see, new Appendix A1 and Fig.AR8a,b,c).

**Reviewer: Please clarify how this is handled regarding the footprints from the LES.**

**Answer:** See the clarification above. We clearly understand and share all the concerns of the Reviewer. The disadvantages of the proposed setup of the numerical experiment were known for the authors at the beginning of this study. The clear and justified method for the footprint determination in LES up to a limited distance $x - x_M$ will be the appropriate restriction of the particle flight. Nevertheless, we choose this setup deliberately as a way for the direct comparison of statistics obtained by Eulerian and Lagrangian methods. For example, this way permits to compare Schmidt numbers, variances, vertical turbulent fluxes (the resolved and the parameterized separately). All of this give additional possibilities for the LES model validation and development of the optimized procedures for the particles transport in LES.

**(3) Normalisation of footprints:**

**Reviewer: On p. 13, l. 14, it is shown that the integral over the footprint function is normalised to one.**

**Answer:** The normalization of footprints was made only for the Fig.3 (AR3) and Fig.5 (AR5), when the different approaches for the subgrid modeling of particles motions were studied. As all curves in these Figures were normalized identically, the comparison is objective. Besides, while the horizontal particle flight was not restricted, the footprint functions, defined by the Eq. (1) and Eq. (2) and the normalized footprints shown in these figures differ by the multiplier $a = F_s(0)/F_s(z_M)$ (here, $F_s$ is the vertical concentration flux of the particles). The total vertical concentration fluxes are nearly independent from the model, so the only impact of normalization is the scaling of the axis $y$ in Fig. 3 (AR3) and Fig. 5 (AR5) (Fig. 4 (AR4) and Fig. 6 (AR6) in revised paper). This does not influence both the results and the conclusions.

Nevertheless, it was mistake by the authors to include the figures with the different normalization into one paper (figures with the different normalization in one paper but not the curves with different normalization in one figure, as it could have been misunderstood due to the unclear

presentation in the former version of the paper).

**Corrections:** We recalculated all the curves, shown in Fig.3 and Fig.5 and removed the sentence concerning normalization from the text. All the conclusions and the descriptions of the results remain unchanged, as well as other footprints functions shown in the paper (e.g., Fig. 13 (AR2 ) in the new version coincides with Fig.11 from the original version of the paper).

**Reviewer: Does this include negative footprint values, too? Or are these treated separately as mentioned on p. 15?**

**Answer:** Yes, negative values were also included.

**Reviewer: Please clarify. The absolute values of the footprint function and hence the cumulated footprint will depend on how negative values are treated. Observed differences in the absolute footprint function values for different footprint approaches (cf. Fig. 11) may be partly due to differences in normalisation procedures.**

**Answer:** Figure 11 was shown without normalization and remains unchanged. The differences are essential.

**Reviewer: Also, is there a threshold value for the distance from the measurement up to where footprint values are considered? The 'flat' trend of the cumulative footprint values suggests that the footprint function would only completely diminish at very large distances from the measurement location. If a threshold value is set, again the selected value will have an impact on the normalisation and the absolute value of the footprint function. Please provide more information on the applied procedure.**

**Answer:** See the previous answer. The threshold value for the collection of footprints was selected large enough to include all the particles (see description of the footprint function grid in the answer to Reviewer: #1).

**(4) Kolmogorov's constant for the structure function in the Inertial Subrange [C0]:**

**Reviewer: First of all, this constant is referred to as 'Kolmogorov constant', a name**

that is usually associated with that in the energy spectra in the Inertial Subrange (and has a value of approximately 1.5).

**Answer:** This constant, as well as the formula $E(k) = C_K \epsilon^{2/3} k^{-5/3}$ appeared for the first time in the paper by Obukhov, 1941 ([2], in Russian) which was published a little bit later than the famous paper by Kolmogorov, 1941 ([1], in Russian), where the equivalent form of this law for the velocity structure function $\langle v^2(r) \rangle \sim \epsilon^{2/3} r^{2/3}$ was discovered. In turn, the Lagrangian velocity structure function $\langle (v(t+\tau) - v(t))^2 \rangle = C_0 \epsilon \tau$ was introduced in Landau and Lifshitz, 1944 ([3], first edition, in Russian) and later independently in [Obukhov A. M., 1959].

From this historical point of view both of these constants $C_K$ and $C_0$ have the equal right to be called the "The Kolmogorov constant" - both of them were introduced first in the papers or the books of other authors and both of them were related to Kolmogorov's (1941) theory.

Although $C_0$ for LSMs is very often referred as the 'Kolmogorov constant' and the dissipation rate $\epsilon$ stands as the single determining parameter for the generative terms in LSMs, we agree with this comment. In practice, the constant $C_0$ in LSMs of ABL is not connected directly with the motions in the inertial subrange and is responsible for the scales outside the range of isotropy.

**Corrections:** Accordingly, we replaced naming 'Kolmogorov constant $C_0$' by the 'parameter $C_0$' or 'value of $C_0$' everywhere except page 7, line 4, where this constant is related to the Lagrangian velocity structure function in the inertial subrange.
In the Conclusions the sentence concerning the constant $C_0$ was extended as follows:

The optimal value for the parameter $C_0$ for LSMs is found to be close to 6 under the conditions considered here. This value coincides with the estimation of Kolmogorov Lagrangian constant in isotropic homogeneous turbulence. It provides additional justification for use of LSMs in stable ABL, due extending their of its applicability over a wider range of scales including the inertial subrange. Stochastic models that use smaller values $C_0 \approx 3 - 4$ (this choice is widespread now) may produce extra mixing and the shorter footprints, respectively. Note that the estimation $C_0 = 6$ is based on the LES results combined with the SHEBA data [Grachev et al., 2013], where the nondimensional vertical velocity RMS was evaluated as $\tilde{\sigma}_w \approx 1.33$ (the exact estimation of this value in LES is restricted by the resolution requirements). In the cases when LSMs utilize smaller values of $\tilde{\sigma}_w$ the parameter $C_0$ should be reduced accordingly (for example, $C_0 \approx 4.7$ will be the best suited parameter for LSMs with the widely used value $\tilde{\sigma}_w \approx 1.25$ prescribed).

**Reviewer: The authors discuss the range of proposed values in the literature, and it is felt that i) the paper by Rizza et al. (2010) might be a valuable addition to the discussion of possible values in the PBL**

**Answer:** We agree. This reference was added with the appropriate commenting (see next answer).

**Reviewer: ii) the employed value in the LES subfilter correction (Eqs. 28 ff) should be provided.**

**Corrections:** We provide the value $C_0 = 6$ in the revised paper, see page 11, line 10.
Besides, the constant $C_K$ was also not specified in the paper. We clarified the procedure of the evaluation of subgrid energy by its extension on non-equidistant grids in accordance with the formulas employed in LES code (p. 11 l. 21-26).

**Reviewer: However, in the present paper it is demonstrated that the results of the LSMs (and in particular LSMT) are sensitive to the choice for $C_0$ – which is per se not particularly new (see, e.g., Rotach et al. (1996) who have sought the 'optimal' value based on comparison to water tank (dispersion) measurements of Willis and Deardorff – and many others, such as Du et al. (1995), Reynolds (1998), as cited in Rizza et al).**

**Answer:** We completely agree with the Reviewer.

**Corrections:**

1) The text beginning from line 13 p, 7 in original version was rewritten as follows (p.7 l.23 - p.8 l. 8):

There is no consensus on the value of $C_0$ as well. Formally, $C_0$ has the meaning of a universal Kolmogorov constant in Eq. (11). The estimation of this constant for an isotropic turbulence using the data of laboratory measurements and DNS provides an interval $C_0 = 6. \pm 0.5$ (see, [Lien and D'Asaro, 2002]). However, the values $C_0 \sim 3 - 4$ are often used for LSM of particle transport in ABL independently from the type of the stratification. These values have been obtained by the different methods. For instance, the value $C_0 = 3.1$ for a one-dimensional LSM corresponds to a calibration performed in [Wilson et al., 1981] according to observation data [Barad, 1958, Haugen, 1959]. This calibration (see, [Wilson, 2015]) assumes that the turbulent

Schmidt number $Sc = K_m/K_s = 0.64$ near the surface (here $K_m$ is the eddy viscosity). It is known that determination of the turbulent Prandtl number $Pr = K_m/K_h$ ($K_h$ - heat transfer eddy diffusivity) and Schmidt number based on observation data is complicated by large statistical errors associated with the problem of self-correlation [Anderson, 2009, Grachev et al., 2007]. Therefore, this method of estimation of $C_0$ cannot be considered as final and should be confirmed by future studies. In [Rizza et al., 2010] the values of $C_0$ were determined using the LES-based evaluations of the velocity structure functions and the Lagrangian spectra in convective and neutrally-stratified ABLs. In this study the LES model had relatively low resolution, which can be insufficient for accurate determination of this constant in the inertial subrange (see discussion on the resolution requirements in [Lien and D'Asaro, 2002]). Nevertheless, the value $C_0 \sim 3$, in the paper by [Rizza et al., 2010] is relevant for LSMs applied to the convective ABL, in that case the constant is also responsible for the energy containing time scales which are well resolved in LES. The detailed overview of the methods of determination of the constant $C_0$ can be found in [Poggi et al., 2008], where the discussion on the disagreements of the different approaches is also included. The results of the LSMs are very sensitive to the choice for $C_0$ as it was shown earlier by [Du et al., 1995, Rotach et al., 1996, Wilson, 2015] and many others. Below we show that the commonly used value of $C_0 \sim 3-4$ can be greatly underestimated for LSMs applied to the stably stratified ABL.

2) We excluded the sentence concerning the value of $C_0$ in the Abstract.

**Reviewer: If indeed the LES were fully validated and all the choices substantiated (see major comment 1), the present simulations would correspond to 'one more tessera' in the picture of a possible nonuniversality of C0, be it due to stability dependence or employed time scales (outside those corresponding to the Inertial Subrange). It is felt that the conclusions drawn in the present paper (one 'case' – even with three heights) do not warrant the quite general conclusions drawn (p. 21, l. 18), i.e. 'the optimal value is found to be close to 6'**

**Answer:** We agree with the Reviewer.

**Corrections:** *We add next clarification:*

The optimal value for the parameter $C_0$ for LSMs is found to be close to 6 under the conditions considered here.

**(5) Footprints plotted in Fig. 11:**

**Reviewer: The footprints plotted in Fig. 11 of the manuscript and listed as Kljun et al. (2015) do not coincide with FFP model results. Plotted below are footprints derived from FFP for the input values mentioned in the manuscript, and optimised parameters for neutral and stable conditions as listed in Kljun et al. (2015). (Note: using the universal FFP parameters, e.g. from the online footprint tool still results in different footprints than those plotted in Fig. 11). It can be seen in Fig. R1 that the peak location of FFP fits very well the peak of LSMT with C0=3 in Fig. 11. Footprint peak values, however, do differ, especially for larger measurement heights. Regarding the absolute values of these peak values please see major concerns (2 and 3) above.**

**Answer:** See answer to this comment above.

**Reviewer: Also, the model of Kljun et al. (2004) is outdated; issues in stable conditions were known and were one of the reasons for the update to the model of Kljun et al. (2015).**

**Answer:** We leave the decision concerning the model of Kljun et al. (2004) up to Reviewer and Editor. This model is available online http://footprint.kljun.net/m2004/varinput.php without notice for caution, so we have used this tool.

**Reviewer: As FFP compares well with the Lagrangian footprint model it is based on (see Fig. R2), and as different settings of $C_0$ produce similar shifts in footprints in LPDM-B (Kljun et al. 2002) and the LPDM used in this study (Fig. R2) - the main question boils down to: what is the 'ultimate truth' and what should a footprint parameterisation be based upon? (See comments above.) This is a very important question and I suggest that the authors highlight this fact even more in their manuscript.**

**Answer:** We are confident that the results presented in this paper are accurate for the purpose of footprint evaluation (see answers to comments above) and will not change substantially if any other LES model with sufficiently good resolution will be used.

The questions remain:

Are the conditions of the numerical setup of the experiment GABLS-1 characteristic for the stable ABL in nature and is this case appropriate for making general conclusions?
We believe that it is true because:

i. This setup is based on the observation data [Kosoviĉ and Curry, 2000] and LES reproduces this case quite well.

ii. Usual nondimensional functions of the similarity theory are well satisfied in this case (see e.g. [Basu and Porté-Agel, 2006, Beare et al., 2006, Glazunov, 2014, Zhou and Chow, 2011] ).

iii. A lot of single column models were tested in similar conditions [Cuxart et al., 2006] and the results of their comparison with the LES were treated as the indicator of models performance under stable stratification.

iiii. Similarity of the turbulent stable ABLs permits to conclude that the results obtained in one case can be generalized for many others.

The fact that FPP predicts footprints based on LPDM-B indicates that it is able to reproduce the correct form of footprint function, that the scaling approach proposed by [Kljun et al., 2015] is well justified and that FPP is able to be calibrated with respect to this stochastic model and with respect to the postulated nondimensional functions. Most probably, FPP, can be rescaled using the other parameters for LPDM-B or any other data set, including the LES results.

We believe, this paper provides sufficient amount of information, concerning model development techniques and the models evaluation. The investigation of other cases will require development of additional scenarios which should be considered as a separate problem.

The authors would be pleased to work in cooperation with the author of the review if he/she is interested in collaboration. In this case, please contact us directly.

**Minor Comments**

**Reviewer: (a) The term "Analytical footprint model" is commonly used for footprint models based on analytical solutions of the diffusion equation by applying a K-theory approach. This is a distinctly different approach than used in the models of Kljun et al. (2004, 2015). The latter models are footprint parameterisations. Please correct throughout the manuscript.**

**Corrections:** The term "Analytical footprint model" was substituted by "footprint parameterisations"

**Reviewer: (b) p. 2, l. 5: '...commonly, the application of these models is limited by the constant flux approximation': this is not true at least for the Kljun et al. papers cited above.**

**Corrections:** This sentence was modified as follows:

Commonly, the applicability of the analytical models is limited by a "constant flux layer" simplification, assuming that the measurement height $z_M$ is much less than the thickness of the ABL $z_i$.

FPP is referred everywhere in the revised paper as the 'footprint parametirization'.

**Reviewer: (c) p. 5, l. 26: If reference is made to 'the lake', this lake must be introduced beforehand. It is not appropriate to explain in brackets that the author apparently works on a 'lake problem'.**

**Corrections:** The term ' lake' is substituted for more neutral 'inhomogeneous surface' which has no direct association with another topic in which authors are involved.

**Reviewer: (d) p. 8, l. 15: Euclidean: spelling?**

**Answer:** It is not the spelling, it is a mistake. The continuous space is considered at this

stage of description, so it will be better to write:

... reduces to minimization of the functional $\Psi(X) = \int_\Omega \varepsilon_{ij}(\vec{x}) \; \varepsilon_{ij}(\vec{x})d\vec{x}$ where $\Omega$ is the model domain and $\varepsilon_{ij}(\vec{x})$ is the the residual of the overdefined system of equations ...

It was corrected.

**Reviewer: (e) According to Eq. 2, fys corresponds to the crosswind-integrated footprint. Please use this well established term rather than 'crosswind averaged footprint' (e.g. p. 14, l. 3 or p. 20, l. 18). Further, in the captions of Figs. 9 and 11, the graphs are referred to as "One-dimensional footprints fys". This would suggest that the footprint at y=0 is plotted. Please clarify.**

**Corrections:** It was corrected.

**Reviewer: (f) Wind profile: from Figure 1, it seems that simulated wind speeds in the surface layer part of the domain are smaller than the 'standard' wind profile for stable conditions (e.g., Stull, 1988; Högström, 1996). Please add a couple of sentences to explain why.**

**Answer:** See the first answer.

**Summary**

In this table we summarise shortly all the comments which were accepted with the following revision of the paper or rejected with the following minor corrections and the justification if it is needed.

| # | Comment | Answer and corrections |
|---|---------|------------------------|
| **Major comments** | | |
| **(1)** | Model validation and argumentation of approaches | Accepted partially. Confirmed by the adding of new results. Some clarifications were included. |
| **(2)** | Absorption condition | Rejected. New confirmations were added. |
| **(3)** | Normalisation of footprints | Accepted partially. Corrected with the minor revision. Main results remain to be unchanged. |
| **(4)** | Kolmogorov's constant for the structure function in the Inertial Subrange [C0] | Accepted. Corrected using the exclusion of too general conclusions and with the correction of the terminology. |
| **(5)** | Footprints plotted in Fig. 11 | Rejected. Minor clarification was inserted. |
| **Minor comments** | | |
| **(a)** | The terminology | Accepted. The appropriate corrections were included. |
| **(b)** | Mistake then citing | Accepted and corrected. |
| **(c)** | Embedded advertising (the use of the word 'lake') | Accepted and excluded. |
| **(d)** | Spelling? | Accepted by the other reason than spelling. Improved. |
| **(e)** | The terminology | Accepted. Appropriate corrections were included. |
| **(f)** | Correctness of the LES results (wind profile) | Rejected. The confirmation was included. |

Landau L.D., Lifshitz E.M.: Mechanics of Continua. OGIZ Gostehizdat, 1944. (in Russian)

[revised manuscript text omitted]

**Response to Editor minor comments**

Authors are grateful for a great job which was done.

Proposed by the authors and recommended by the Editor minor revision of the manuscript was performed.

Next clarification was added in Conclusions (p.25, l. 14-19): "We emphasize that a very simple case of the moderately stratified stable ABL in almost steady state conditions was considered here. This setup of numerical experiments permits the detailed intercomparison of a different approaches for the particle dispersion modeling which utilize identical simplifications. On the other hand, in real environment the scalar flux footprint functions can be greatly influenced by the meteorological non-stationarity, the peculiarities of mixing inside the roughness layer, internal radiative heating or cooling in ABL and so on. Also, a wider investigation of different stability regimes from neutrality to strong stratification must be undertaken in future studies to confirm the universality of the findings."

Besides, we have added two curves in Fig.5 (the results, obtained in LES with subgrid Lagrangian RDM) and the appropriate short description of these curves in Sect.4.2.4 (p.18, l.1-4) .

[revised manuscript text omitted]

Figure AR9.2 (Fig.S1.1 from the Supplements to the paper) Crosswind integrated concentration $\tilde{C}^y = C^y U z_i / Q$ depending on normalized height $z/z_i$ and non-dimensional distance from the source $X = xw^*/(Uz_i)$. (a) CWIC profiles $\tilde{C}^y$ computed in LES with different resolution (solid lines) in comparison with laboratory data (squares). (b) CWIC isolines computed with grid steps $\Delta_g = 10$ m$\approx z_i/100$ and $\Delta_g = 40$ m$\approx z_i/25$ (dashed line - first computational level $z = \Delta_g$).

[Figure]

Figure AR10: **(Fig.S1.3 from the Supplements to the paper)** Footprints $f_s^y$ (a,b) and cumulative footprints $F$ (c,d) for the sensor heights $z_M$=10m (a,b) and $z_M$=100m (c,d), computed with the different spatial resolution in LES. Symbols - observational data [Leclerc et al., 1997]

[Figure]

Figure AR11: Modified Fig.3. Concentration vertical flux profile obtained in LES with the absorbtion condition applied at z=100 m and the liner flux profile as it was predicted by Stull, 1988 or Beare et al., 2006 (blue dashed straight line)